palaeontology/evolution

Clupeomorpha, computed tomography, ecological release, ichthyology, Palaeogene, piscivory

# Large-bodied sabre-toothed anchovies reveal unanticipated ecological diversity in early Palaeogene teleosts

Alessio Capobianco[1,2], Hermione T. Beckett[3,4], Etienne Steurbaut[5,6], Philip D. Gingerich[1,2], Giorgio Carnevale[7] and Matt Friedman[1,2]

[1]Department of Earth and Environmental Sciences, University of Michigan, Ann Arbor, MI, USA
[2]Museum of Paleontology, University of Michigan, Ann Arbor, MI, USA
[3]Department of Earth Sciences, University of Oxford, Oxford, UK
[4]Department of Biology, King's High School for Girls, Warwick, UK
[5]Royal Belgian Institute of Natural Sciences, Brussels, Belgium
[6]Department of Earth and Environmental Sciences, KU Leuven, Leuven, Belgium
[7]Dipartimento di Scienze della Terra, Università degli Studi di Torino, Torino, Italy

AC, 0000-0002-6096-3875; HTB, 0000-0003-4475-021X; PDG, 0000-0002-1550-2674; GC, 0000-0002-3433-4127; MF, 0000-0002-0114-7384

**Author for correspondence:**
Alessio Capobianco
e-mail: acapo@umich.edu

Many modern groups of marine fishes first appear in the fossil record during the early Palaeogene (66–40 Ma), including iconic predatory lineages of spiny-rayed fishes that appear to have originated in response to ecological roles left empty after the Cretaceous/Palaeogene extinction. The hypothesis of extinction-mediated ecological release likewise predicts that other fish groups have adopted novel predatory ecologies. Here, we report remarkable trophic innovation in early Palaeogene clupeiforms (herrings and allies), a group whose modern representatives are generally small-bodied planktivores. Two forms, the early Eocene (Ypresian) †*Clupeopsis* from Belgium and a new genus from the middle Eocene (Lutetian) of Pakistan, bear conspicuous features indicative of predatory ecology, including large size, long gapes and caniniform dentition. Most remarkable is the presence of a single, massive vomerine fang offset from the midline in both. Numerous features of the neurocranium, suspensorium and branchial skeleton place these taxa on the engraulid (anchovy) stem as the earliest known representatives of the clade. The identification of large-bodied, piscivorous anchovies contributes to an

emerging picture of a phylogenetically diverse guild of predatory ray-finned fishes in early Palaeogene marine settings, which include completely extinct lineages alongside members of modern marine groups and taxa that are today restricted to freshwater or deep-sea environments.

# 1. Introduction

Body fossils [1–3], otoliths [4], ichthyoliths [5,6] and molecular clocks [7,8] point to the early Palaeogene as a time of remarkable diversification and innovation among marine fishes. Most of the groups familiar from modern marine ecosystems appeared during this interval, along with their distinctive morphological adaptations. So striking is this pattern that the evolution of marine teleosts after the Eocene has been described by some as 'mere tinkering' [1]. One of the clearest impacts of the Cretaceous/Palaeogene (K/Pg) extinction on teleosts was the extinction of large predatory taxa [9–11], which has been implicated in permitting the subsequent diversification of lineages like mackerels and kin (Scombridae), barracudas (Sphyraenidae), billfishes (Xiphiodei) and jacks (Carangidae) that first appear near the Palaeocene–Eocene boundary [3,8,12,13]. All of these examples belong to a group of fishes called acanthomorphs, or spiny-rayed teleosts, which represent the dominant fish group in marine settings since the beginning of the Cenozoic [1,2]. However, the hypothesis that opportunity arising from the K/Pg extinctions fuelled diversification predicts that other groups may have also experimented with new roles in the early Palaeogene, although this has been little investigated (but see [14,15]) despite its significance for understanding the structure of marine faunas at that time.

Here we report a new early–middle Eocene clade of large-bodied clupeiform (herrings and anchovies) fishes, the anatomy of which has been revealed by micro-computed tomography (μCT). This group, represented by the previously described †*Clupeopsis straeleni* from the Ypresian of Belgium [16] and a new genus and species from the Lutetian of Pakistan, is characterized by remarkable dental specializations: a single row of enlarged dentary teeth combined with a single massive vomerine fang that extends to the ventral margin of the mandibular symphysis. These fossils force a reconsideration of trophic diversity among marine clupeiforms in the early Palaeogene, which are otherwise represented by small-bodied, probable planktivores apparently similar to the vast majority of living clupeiforms [17–20]. More broadly, they point to previously unappreciated trophic innovation in an early Palaeogene marine setting that has not persisted to the modern day.

# 2. Material and methods

## 2.1. Micro-computed tomography (μCT) scanning

The holotypes of †*Clupeopsis straeleni* (MRHNB IG 8630) and GSP-UM 37, as well as representative examples of extant clupeiforms, were imaged using Nikon XT H 225ST industrial μCT scanners at the University of Michigan and the Natural History Museum, London. Reconstructed datasets were visualized and segmented using Mimics v. 19.0 (Materialise, Belgium). Models of segmented skeletal elements were exported as surface files (.ply) and rendered as high-quality images in Blender v. 2.79 (blender.org). Individual scanning parameters of new tomograms generated for this study are provided in the electronic supplementary material.

## 2.2. Comparative material

μCT scans obtained from the following formalin-fixed specimens of extant clupeiform species were examined as comparative material.

Clupeidae. *Clupea harengus* UF 184063 (Morphosource media M44470), *Odaxothrissa mento* UMMZ 195016.

Pristigasteridae. *Ilisha elongata* UF 143661 (Morphosource media M44747), *Odontognathus mucronatus* UF 135948 (Morphosource media M44474).

Chirocentridae. *Chirocentrus dorab* UMMZ 238306, *Chirocentrus nudus* UMMZ 213502.

Engraulidae. *Lycengraulis grossidens* UMMZ 143053, *Setipinna* [*Lycothrissa*] *crocodilus* UMMZ 209911.

In addition to the material listed here, further observations of clupeiform osteology were derived from descriptive accounts [21–26]. Following best practices in the accessibility of tomographic data

[27], we have made tomograms, .mcs files and .ply files of segmented structures available at the Dryad Digital Repository.

## 2.3. Institutional abbreviations

GSP-UM, Geological Survey of Pakistan, Quetta, Pakistan, specimens collected during joint expeditions with the University of Michigan Museum of Paleontology; MRHNB, Musée Royal d'Histoire Naturelle de Belgique (now RBINS), Brussels, Belgium; RBINS, Royal Belgian Institute of Natural Sciences, Brussels, Belgium; UF, University of Florida, Gainesville, FL, USA; UMMZ, University of Michigan Museum of Zoology, Ann Arbor, MI, USA.

## 2.4. Dagger symbol

Following the convention of [28], the dagger symbol (†) precedes extinct groups.

# 3. Results

**Systematic palaeontology.**
 Teleostei Müller, 1845 [29]
 Clupeomorpha Greenwood, Rosen, Weitzman and Myers, 1966 [30]
 Clupeiformes Bleeker, 1859 [31] *sensu* Grande, 1985 [21]
 Clupeoidei Jordan, 1923 [32] *sensu* Greenwood, Rosen, Weitzman and Myers, 1966 [30]
 Engrauloidea Grande, 1985 [21]
 †*Clupeopsis straeleni* Casier, 1946 [16]

## 3.1. Material

MRHNB IG 8630 (holotype no. 276), Royal Belgian Institute of Natural Sciences, Brussels, Belgium. Holotype and only known specimen, representing a three-dimensionally preserved individual comprising an almost complete skull plus incomplete body extending to the level of the dorsal fin (figure 1; electronic supplementary material, figures S1–S6). The specimen measures 278 mm from the tip of the snout to the broken posterior end of the body.

## 3.2. Locality and horizon

Dubois clay pit, Chièvres, Hainaut, Belgium, 3.7 m above Ypresian basal gravel [33]. This represents the basal part of the Orchies Clay Member, providing an age constraint of approximately 54.40–54.05 Ma. The Orchies Clay Member is interpreted as being deposited in an outer neritic marine setting, but its base appears to represent a shallower facies [34]. Additional details are provided in the electronic supplementary material.

## 3.3. Diagnosis

Clupeiform with triangular-shaped skull in lateral and dorsal views; vomer bearing two large tooth pits; single vomerine fang, extending ventrally beyond the mandibular symphysis when jaws closed and representing the largest tooth; dorsolaterally oriented pre-epiotic fossa; robust and slightly curved toothless maxilla measuring approximately 75% of neurocranium length; straight and robust dentary bearing caniniform teeth throughout its length; largest dentary tooth approximately 20% of length of orbital cavity; round, greatly expanded posterior infraorbital covering most of hyomandibula and quadrate and part of the maxilla; hyomandibula slightly reclined posteriorly; small lateral horizontal lamina of the ectopterygoid underlying the orbit; second basibranchial longest element of the basibranchial series.

## 3.4. Remarks

The postcranium and surficial aspects of the skull were described previously [16]. We focus on details of the cranium revealed by external restudy and µCT.

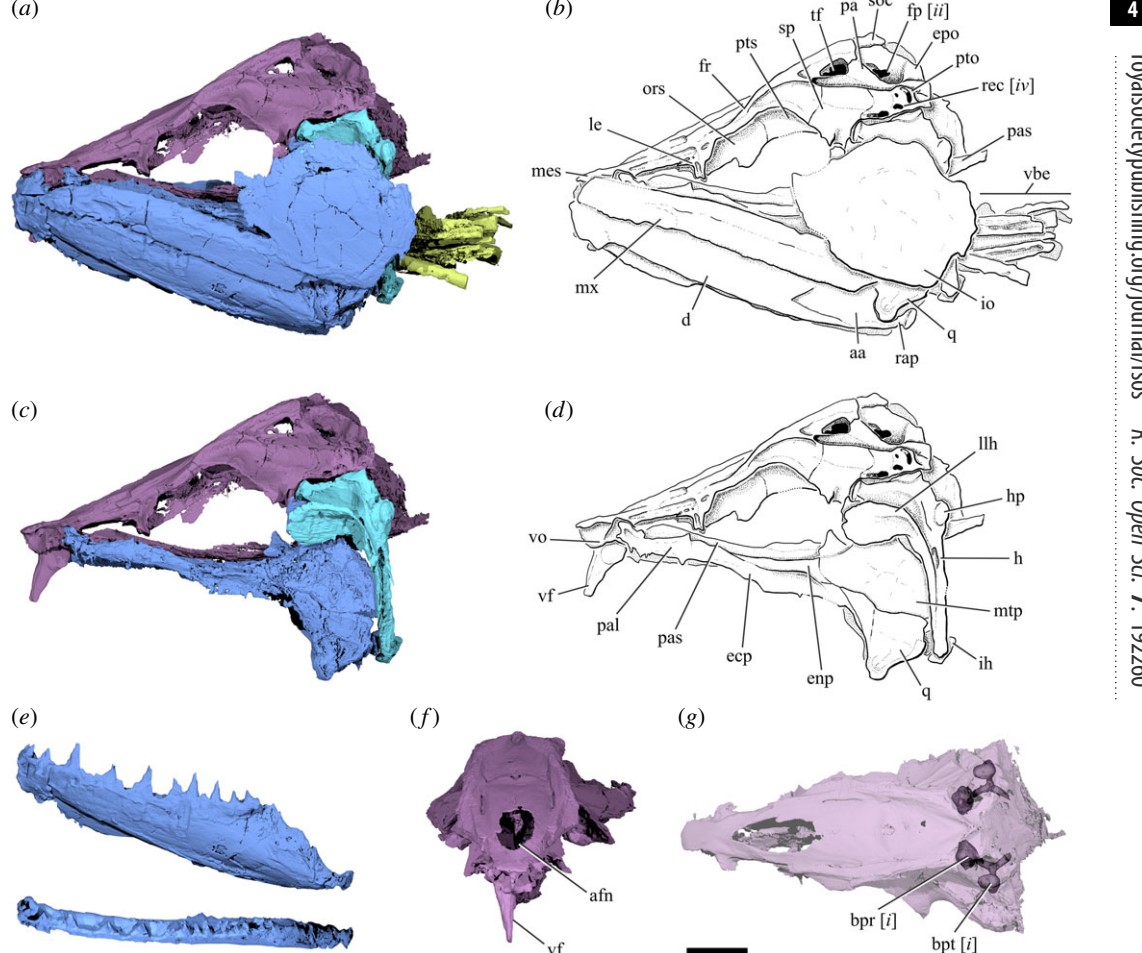

**Figure 1.** Cranial anatomy of †*Clupeopsis straeleni* Casier 1946 (MRHNB IG 8630, holotype). (*a*) Rendering and (*b*) line drawing of skull, excluding the opercular series. (*c*) Rendering and (*d*) line drawing of braincase, palate and suspensorium in left lateral view. (*e*) Renderings of left dentary in lateral (top) and dorsal (bottom) views. (*f*) Rendering of braincase in anterior view, highlighting the vomerine fang. (*g*) Rendering of braincase in dorsal view, with partial transparency to reveal the prootic and pterotic bullae. Colours indicate different cranial regions: neurocranium (purple), jaws, palatoquadrate and cheek bones (blue), dorsal portion of hyoid arch (cyan), branchial skeleton (chartreuse). Roman numerals in italics refer to synapomorphic characters within Clupeomorpha explained in the Discussion text. Scale bar: 10 mm. Abbreviations: aa, anguloarticular; afn, anterior frontal fontanelle; bpr, prootic bulla; bpt, pterotic bulla; d, dentary; ecp, ectopterygoid; enp, endopterygoid; epo, epioccipital; fp, pre-epiotic fossa; fr, frontal; h, hyomandibula; hp, hyomandibular process; ih, interhyal; io, infraorbital; le, lateral ethmoid; llh, lateral laminar process of hyomandibula; mes, mesethmoid; mtp, metapterygoid; mx, maxilla; ors, orbitosphenoid; pa, parietal; pal, palatine; pas, parasphenoid; pto, pterotic; pts, pterosphenoid; q, quadrate; rap, retroarticular process; rec, *recessus lateralis*; soc, supraoccipital; sp, sphenotic; tf, temporal foramen; vbe, ventral branchial elements; vf, vomerine fang; vo, vomer.

## 3.5. Description

The skull is triangular in dorsal and lateral view. The preorbital region of the neurocranium is slightly longer than the postorbital region. The frontals are the largest bones of the skull roof; they are narrow and elongated, broadest at mid-length and slightly tapering posteriorly. Two pronounced longitudinal crests extend on the dorsolateral side of the frontals. These ridges start above the orbits and continue posteriorly on the supraoccipital, where they join medially to form an occipital crest. The supraorbital canal is partially exposed medial to the longitudinal crest, in a sulcus above the orbit. A large median fontanelle is present between the anterior portion of the frontals. Posteriorly, the frontal forms the anterior and dorsal margins of the temporal foramen, which is also bound by the parietal. The parietals are completely separated from each other along the midline by the frontals anteriorly and by the supraoccipital posteriorly. The sphenotic is broad, concave and laterally projecting. The pterotic forms the postero-lateral corner of the neurocranium and is dorsoventrally compressed. The lateral region of the pterotic houses the majority of the *recessus lateralis* chamber, an intracranial space where

several cephalic sensory canals converge. The pterotic bears four distinct openings for the *recessus lateralis* (figure 1; electronic supplementary material, figure S2). The medial region of the pterotic houses a pterotic bulla that is connected with a larger prootic bulla; together they form part of the otophysic connection. The posterior part of the frontal, the parietal and the pterotic form a prominent post-temporal groove. The pterotic, parietal and epioccipital delimit a relatively large pre-epiotic fossa with a dorsolateral opening slightly smaller than the temporal fenestra.

The dorsoventrally flattened mesethmoid extends a short distance anterior to the vomer. The irregularly shaped lateral ethmoid forms the anterior margin of the orbit. The vomer houses two very large tooth sockets, of which only the right is occupied (electronic supplementary material, figure S3). It bears a massive fang that projects ventrally beyond the mandibular symphysis. The vomerine fang is laterally compressed and slightly curved posteriorly. The toothless parasphenoid is elongated, dorsoventrally compressed and slightly arched. Below the otic region of the braincase, it divides into two slender branches that extend posteriorly beyond the occipital condyle.

The jaws extend posteriorly beyond the orbit. The maxilla is toothless, relatively deep and robust. Even though the maxilla is completely devoid of teeth, its ventral margin appears to bear micro-serrations along most of its length. We were not able to identify any premaxillae or supramaxillae. Fragments of bone antero-medial to the maxilla might represent parts of the premaxilla. The mandible includes both a dentary and an anguloarticular; the presence of a retroarticular cannot be determined. The dorsal margin of the dentary bears a row of 11–12 caniniform teeth. The first two teeth are smaller than the others and positioned at the anterior tip of the lower jaw. Some teeth are separated from each other by pits. The coronoid process is not strongly developed. A low retroarticular process is present on the anguloarticular.

The hyomandibula is directed obliquely, with its distal tip posterior to its articular head. Its ventral arm extends to the level of the quadrate, while its dorsal head bears an extensive lateral laminar process that overlaps the metapterygoid. The interhyal is small and cylindrical. The quadrate bears a large laminar outgrowth at its anterior margin and an anteriorly directed articular condyle at its ventral corner. The quadrate almost completely obscures the large, wedge-shaped symplectic in lateral view (electronic supplementary material, figure S4). The anterodorsal margin of the metapterygoid bears a deep notch. The ectopterygoid and palatine together form the slender anterior ramus of the palate. Poorly resolved projections on the ventral margin of the palatine might represent teeth. The palatine presents a posteriorly directed flat surface that, together with the lateral ethmoid that directly overlies it, forms the anterior margin of the orbit. The ectopterygoid bears a small lateral horizontal lamina under the orbital region (electronic supplementary material, figure S5a). The endopterygoid is plate-like and lies in a horizontal plane.

The gill skeleton is partially articulated (electronic supplementary material, figure S6), but the identity and morphology of individual elements is difficult to establish due to poor contrast in tomograms. Dorsal and ventral hypohyals are clearly separate and articulate posteriorly with the anterior ceratohyals. There is no trace of a basihyal nor of a basihyal toothplate. The basibranchial toothplate appears to be fused only to the second basibranchial. The second basibranchial is the longest of the basibranchial series.

Only two infraorbitals are identifiable. The first, representing either the first or second infraorbital, is elongated and slightly curved and lies ventral to the anterior portion of the orbit (electronic supplementary material, figure S2). The second, which probably represents the third infraorbital based on its position, is rounded and greatly expanded, covering most of the hyomandibula, quadrate and the posterior region of the maxilla. The infraorbital sensory canal extends through the anterior portion of the third infraorbital. Only discontinuous segments of the canal can be identified in the anterior elongated infraorbital. One scleral ossicle can be identified on the left side of the specimen.

While the opercular series is present in the specimen (electronic supplementary material, figure S1), individual bones are difficult to identify in the tomograms due to poor contrast. The preopercle has a gently curved anterior margin, with no clear distinction between dorsal and ventral limbs. The opercle appears to be relatively large, with length roughly equal to its depth.

†*Monosmilus chureloides* gen. et sp. nov.

## 3.6 Etymology

Generic name from the combination of the Ancient Greek *mónos* (single) and *smil'e* (knife), referring to the single massive vomerine fang. Specific name from the combination of *Churel*, the name in Urdu of a shapeshifting vampire-like demon with large fangs or tusks, with the suffix *-oides*, indicating similarity.

**Figure 2.** Cranial anatomy of †*Monosmilus chureloides* gen. et sp. nov. (GSP-UM 37, holotype). (*a*) Rendering and (*b*) line drawing of specimen in left lateral view. (*c*) Rendering of basibranchial series with hypohyals (dark green) in left lateral view. (*d*) Rendering of braincase in anterior view, highlighting the vomerine fang. (*e*) Rendering of right dentary (incomplete) in medial view, with loose tooth crowns (probably replacement teeth) highlighted in red. Colours indicate different cranial regions: neurocranium (purple), jaws, palatoquadrate and cheek bones (blue), dorsal portion of hyoid arch (cyan), branchial skeleton (chartreuse). Scale bar: 10 mm. Abbreviations: afn, anterior frontal fontanelle; bb1, first basibranchial; bb2, second basibranchial; bb3, third basibranchial; bbtp, basibranchial toothplate; d, dentary; dh, dorsal hypohyal; f, frontal; ecp, ectopterygoid; enp, endopterygoid; h, hyomandibula; io, infraorbital; le, lateral ethmoid; mpt, metapterygoid; mx, maxilla; ors, orbitosphenoid; pal, palatine; pas, parasphenoid; q, quadrate; sp, sphenotic; sr, scleral ring; vf, vomerine fang; vh, ventral hypohyal; vo, vomer.

## 3.7. Material

GSP-UM 37, Geological Survey of Pakistan, Quetta, Pakistan. Holotype and only known specimen, an incomplete but three-dimensionally preserved skull broken anteriorly at the tip of the snout and posteriorly in advance of the occipital condyle (figure 2; electronic supplementary material, figures S5, S7–10). The specimen measures 104 mm between these broken surfaces. It was collected during a November–December 1977 field season of the Geological Survey of Pakistan and the University of Michigan Museum of Paleontology [35].

## 3.8. Locality and horizon

Rakhi Nala (locality RN-4) on the east side of the Sulaiman Range in the Dera Ghazi Khan District of western Punjab Province, Pakistan. The 'lower chocolate clays' of the Domanda Formation (electronic supplementary material, figure S7) were deposited in a shallow coastal marine environment [35]. Nannoplankton stratigraphy (zone NP 15) constrains the Domanda Formation to the Lutetian stage/ age of the early–middle Eocene [36].

## 3.9. Diagnosis

Clupeiform with vomer bearing two large tooth pits; single vomerine fang representing the largest tooth; parasphenoid straight in lateral view; orbitosphenoid antero-ventrally contacting the parasphenoid; occipital region posteriorly elongated; robust and straight maxilla lacking teeth; dentary bearing greatly enlarged, postero-medially recurved caniniform teeth throughout its length; largest dentary tooth approximately 70% of length of orbital cavity; bulbous lateral ethmoids; anterior part of palatine triangular in lateral view; horizontal lamina of the ectopterygoid under the orbit and directly overlying the maxilla; third basibranchial longest element of the basibranchial series.

## 3.10. Notes

The attribution of two closely related species possibly forming a monophyletic group to two separate genera is subjective. †*Monosmilus chureloides* differs from †*Clupeopsis straeleni* in several morphological

features, including: proportionally larger teeth on vomer and dentary; bulbous (rather than irregularly flattened) lateral ethmoid; postorbital region of the neurocranium longer (instead of shorter) than preorbital region; contact between orbitosphenoid and parasphenoid; larger lateral horizontal lamina of the ectopterygoid; longer and broader endopterygoid; third (instead of second) basibranchial as the longest element of the basibranchial series. We consider these differences sufficient to justify the erection of a new genus for †M. chureloides.

## 3.11. Description

The dorsal part of the neurocranium is incomplete, with only the eroded mesethmoid, frontals and left sphenotic present in the orbital and ethmoid regions. The posterodorsal part of the neurocranium is missing, along with the occiput.

The narrow, elongate vomer underlies the ethmoid region, and terminates at a narrow point in the orbital region. It bears two large, deep pits (electronic supplementary material, figure S9), of which only the left is occupied by a massive, laterally compressed fang. The vomerine fang bears a weakly developed keel along its posterior margin. Its distal tip is broken, so its extent is not clear. The lateral ethmoids are large and bulbous and define the anterior margin of the orbits. The parasphenoid is almost straight in lateral view, only curving gently dorsally in the occipital region. It is plank-like, dorsoventrally compressed in the orbital region. Below the otic region it splits into a pair of deep and laterally compressed processes, which extend at least back to the broken posterior margin of the specimen. The foramen for the internal carotid pierces the parasphenoid immediately posterior to the orbital region (electronic supplementary material, figure S9). The antero-ventrally expanded orbitosphenoid extends to the parasphenoid in the anteriormost part of the orbital region, forming a partial septum. The incomplete sphenotic bears part of the anterior hyomandibular facet. Posterior to this facet, two large foramina pierce the prootic. The more anterodorsal of these is for the hyomandibular branch of cranial nerve VII (electronic supplementary material, figure S9). Although it is incomplete, the postorbital region of the neurocranium is longer than the preorbital region.

Within the upper jaw, only the maxilla is preserved, albeit incompletely (electronic supplementary material, figure S8). It is straight, relatively robust and devoid of teeth. The lower jaws are fragmentary, represented by partial dentaries bearing at least eight large, caniniform teeth that are slightly recurved postero-medially. These are generally separated from each other by deep pits. Six to seven loose tooth crowns are preserved parallel to the long axis of the jaw. Some of these occupy pits and appear to be replacement teeth that are not yet connected by bone to the underlying jaw. The external lamina of the dentary partially covers the bases of the teeth.

There is no clear separation between palatine and ectopterygoid, which appear to be toothless with the exception of one or two teeth on the ventral margin of the palatine. The anterior portion of the palatine is triangular in lateral view and is bounded postero-medially by a flat and broad surface, which—together with the posterior surface of the lateral ethmoid—forms the anterior margin of the orbit. In the orbital region, the palatine-ectopterygoid is dorsoventrally compressed and bears a lateral horizontal lamina that overlies the maxilla (electronic supplementary material, figure S5). The endopterygoid is toothless and large, bearing two transverse ridges that divide its dorsal surface into three concave fossae. The metapterygoid is incomplete posteriorly but has a strongly concave anterior margin. A broken sliver of bone represents all that remains of the quadrate. The anterior extent of the quadrate is constrained by the dentary, and it is apparent that the jaw joint was located far posteriorly. This position, combined with the more anterior location of the small portion of the hyomandibular head that is preserved, indicates that the hyomandibula was strongly inclined posteriorly.

The dorsal and ventral hypohyals articulate posteriorly with the incompletely preserved and plate-like anterior ceratohyal (electronic supplementary material, figure S10). There is no ossified basihyal or basihyal toothplate, but there is a well-developed basibranchial toothplate that is tightly fused to the second and third—but not first—basibranchials. The second basibranchial is longer than the first and bears long flanges that extend ventrally from its lateral surface. The third basibranchial is the longest, with the toothplate covering only its anteriormost portion. Several additional components of the gill skeleton are preserved, including well-developed hypobranchials and ceratobranchials and part of the dorsal gill skeleton, but individual bones are displaced and difficult to identify (electronic supplementary material, figure S10). Rakers, if present, must have been very small.

Impressions and bone fragments on the outer surface of the specimen represent infraorbitals, but their shape and number cannot be determined. Two scleral ossicles are preserved on the left side and one on the right side of the specimen.

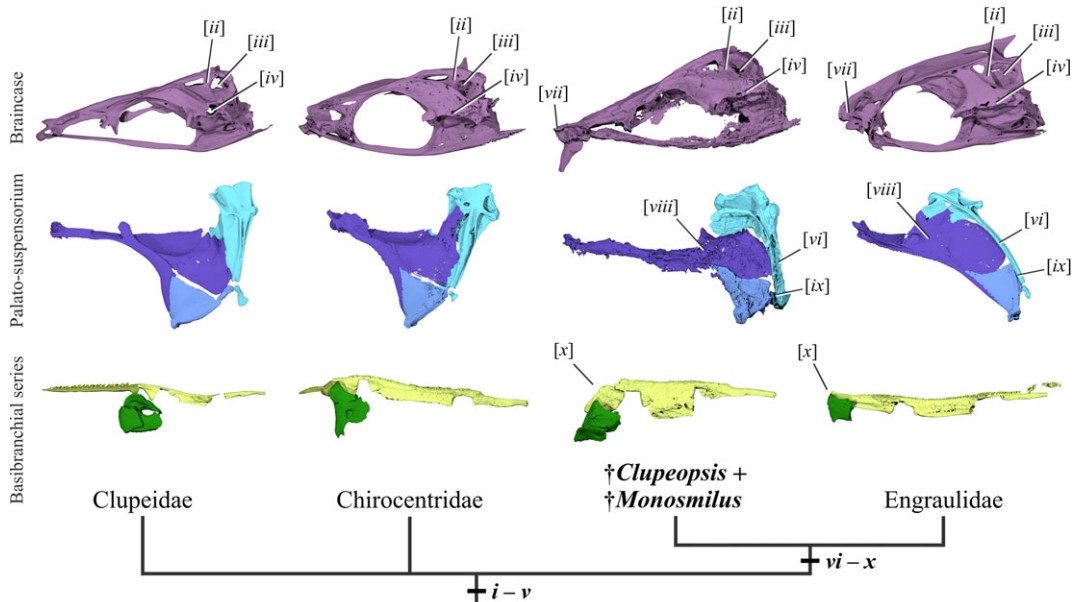

**Figure 3.** Comparative morphology of sabre-toothed stem-anchovies and other clupeiforms. Left lateral views of braincases, palato-suspensoria and basibranchial/basihyal series belonging to representative clupeiforms. From left to right: *Odaxothrissa mento* (Clupeidae), *Chirocentrus dorab* (Chirocentridae), sabre-toothed stem-anchovies (braincase and palato-suspensorium of †*Clupeopsis straeleni*, basibranchial series of †*Monosmilus chureloides*), *Setipinna* [*Lycothrissa*] *crocodilus* (Engraulidae). Synapomorphies of successively restrictive clupeiform sub-clades are indicated in italics Roman numerals and marked over a simplified phylogeny of Clupeoidei, indicating the stem engraulid position of †*Clupeopsis* and †*Monosmilus* and the uncertain relationships at the Clupeoidei base. Details about these synapomorphic characters can be found in the Discussion text. Colours indicate different sets of cranial bones: neurocranium (purple), palate (violet), quadrate and symplectic (blue), hyomandibula and interhyal (cyan), basibranchials (chartreuse), basihyal (ochre green, if present), hypohyals (dark green). Renderings not at same scale.

## 4. Discussion

### 4.1. †*Clupeopsis* and †*Monosmilus* as a monophyletic group

†*Clupeopsis* and †*Monosmilus* are united by a peculiar modification of the vomerine dentition: a pair of laterally adjacent tooth loci, of which only one is occupied at a time by a massively enlarged fang. Expanded vomerine fangs are rare in teleosts, and to our knowledge, no groups share the asymmetric pattern common to these two Eocene taxa. Instead, other arrangements include a single midline fang (e.g. the deep-sea eel *Monognathus* [37,38], the pearlfish *Pyramodon* [39]), a row of midline fangs (e.g. some dysommine cutthroat eels [40], some ophichthid snake eels [41], the Cretaceous aulopiform †*Cimolichthys* [42]), paired fangs (e.g. smelt *Osmerus* [43], some lophiid anglerfishes [44]) or multiple longitudinal rows of fangs (e.g. some serrivomerid sawtooth eels [45] and some ophichthid snake eels [46]). †*Clupeopsis* and †*Monosmilus* bear this unusual vomerine dentition in conjunction with a single row of greatly enlarged dentary fangs in association with reduced dentition on the maxilla. In addition to their specialized dentition, both †*Clupeopsis* and †*Monosmilus* share a lateral horizontal lamina of the ectopterygoid that overlies the maxilla.

### 4.2. †*Clupeopsis* and †*Monosmilus* as clupeoid clupeomorphs

Evidence that †*Clupeopsis* and †*Monosmilus* are clupeomorphs, and more specifically clupeoids, derives from the better-known †*Clupeopsis*. Here and in following sections, small Roman numerals refer to some characters illustrated in figures 1 and 3. †*Clupeopsis* shows the following clupeomorph synapomorphies: (i) pterotic and prootic bulla associated with otophysic connection (character 2 [in part] of [21]) and (ii) a well-defined pre-epiotic fossa (character 5 of [21]). Clupeiform synapomorphies apparent in †*Clupeopsis* are (iii) parietals separated by frontals and supraoccipital (character 10 of [21]) and (iv) a *recessus lateralis* chamber (character 9 of [21]). Finally, †*Clupeopsis* shares with Clupeoidei (v) the absence of a trunk lateral line canal (character 16 of [21]). None of these regions is preserved

in †*Monosmilus*. Instead, the placement of †*Monosmilus* within these groups is based on strong evidence that it is closely related to †*Clupeopsis* (see previous section).

## 4.3. †*Clupeopsis* and †*Monosmilus* as stem engraulids

†*Clupeopsis* and †*Monosmilus* share numerous derived characters with engraulids, but in many cases, these features show a less extreme form of the condition in the Eocene genera than in modern anchovies (figure 3): (vi) suspensorium posteriorly inclined, but less than in modern engraulids (observed in †*Clupeopsis*, inferred in †*Monosmilus*; character 18 [in part] of [21]); (vii) mesethmoid projecting in advance of vomers, but less than in modern engraulids (observed in †*Clupeopsis*, but cannot be checked in †*Monosmilus*; character 19 of [21]); (viii) substantial portion of metapterygoid situated anterodorsal to quadrate (apparent in both genera, but probably linked to suspensorium angle; character 1 of [22]); (ix) ventral limb of hyomandibula meeting the posterior margin of the quadrate (observed in †*Clupeopsis*, but cannot be checked in †*Monosmilus*; character 2 [in part] of [22]) and (x) absence of bony basihyal and basihyal toothplate (observed in †*Monosmilus*, probably in †*Clupeopsis*; characters 1 and 2 of 'Engrauloidae' of [23]). To these established characters, we add three observations that support engraulid affinities. First, both genera share with engraulids a straight maxilla that contrasts strongly with the curved geometry characteristic of other clupeomorphs. Second, like engraulids but unlike other clupeomorphs, †*Clupeopsis* has a greatly reduced coronoid process (condition in †*Monosmilus* cannot be assessed). Third, †*Clupeopsis* shares with engraulids an expansion of the lateral laminar process of the hyomandibula that overlaps part of the metapterygoid (condition in †*Monosmilus* cannot be assessed).

†*Clupeopsis* and †*Monosmilus* retain primitive traits that exclude them from the engraulid crown. These genera lack both fusion between the first basibranchial and the basibranchial toothplate (character 5 [in part] of 'Engrauloidae' of [23]), as well as transverse bony struts enclosing an enlarged supraorbital canal in the frontal [26]. Moreover, †*Clupeopsis* has an anteriorly—rather than posteriorly—directed articular surface of the quadrate (character 18 [in part] of [21]; condition in †*Monosmilus* cannot be assessed).

## 4.4. Implications for clupeiform evolution and ecological diversification

†*Clupeopsis* and †*Monosmilus* are large in comparison to extant anchovies and indeed clupeiforms more generally. The incompleteness of the two specimens precludes exact measurements, but estimates can be made based on proportions in living species. Linear regressions of body length with respect to head length in extant clupeiforms yield body lengths of just below half a metre and one metre for †*Clupeopsis* and †*Monosmilus*, respectively (electronic supplementary material, figure S13). These large sizes in conjunction with well-developed caniniform dentition and slender mandibles consistent with rapid jaw closing suggest a predatory—and probably piscivorous—feeding ecology for †*Clupeopsis* and †*Monosmilus* [47]. The majority of modern clupeiforms are planktivores [48]. Similar ecologies are inferred for most fossil forms [17–19,49] (but see the Early Cretaceous †*Cynoclupea* [50]). However, among extant clupeiforms, piscivory is the second-most common dietary strategy and appears to have evolved multiple times independently from zooplanktivory [48]. Extant piscivorous taxa have a variety of tooth morphologies, including complete absence of teeth, but all clupeiforms with caniniform teeth are piscivorous [51]. These include members of Chirocentridae (*Chirocentus*), Clupeidae (*Odaxothrissa*), Pristigasteridae (*Chirocentrodon bleekerianus*) and Engraulidae (*Setipinna* [*Lycothrissa*] *crocodilus* and *Lycengraulis*) [51]. Among these, only chirocentrids show a development of the caniniform dentition comparable to that of †*Monosmilus* and †*Clupeopsis*.

The relationships among major clupeoid lineages are unclear. Genomic-scale data are available for few species [52], while taxonomically well-sampled molecular phylogenies include only a handful of loci [48,53]. The placement of chirocentrids is especially unclear, with this group resolved in a variety of positions by different molecular analyses [48,53,54]. Di Dario [22] hypothesized a sister-group relationship between engraulids and chirocentrids on the basis of shared features in multiple anatomical systems. Our placement of the large-fanged †*Monosmilus* and †*Clupeopsis* as stem engraulids provides a new perspective bearing on the hypothesized relationship between chirocentrids and anchovies. Although these Eocene genera clearly possess many derived features of engraulids that are absent in chirocentrids, the overall structure of mandibular dentition (in both genera) and braincase (in †*Clupeopsis*; incomplete in †*Monosmilus*) is remarkably similar to that of *Chirocentus*. These features could represent generalized conditions of the putative engraulid/chirocentrid radiation, subsequently lost in more crownward engraulids.

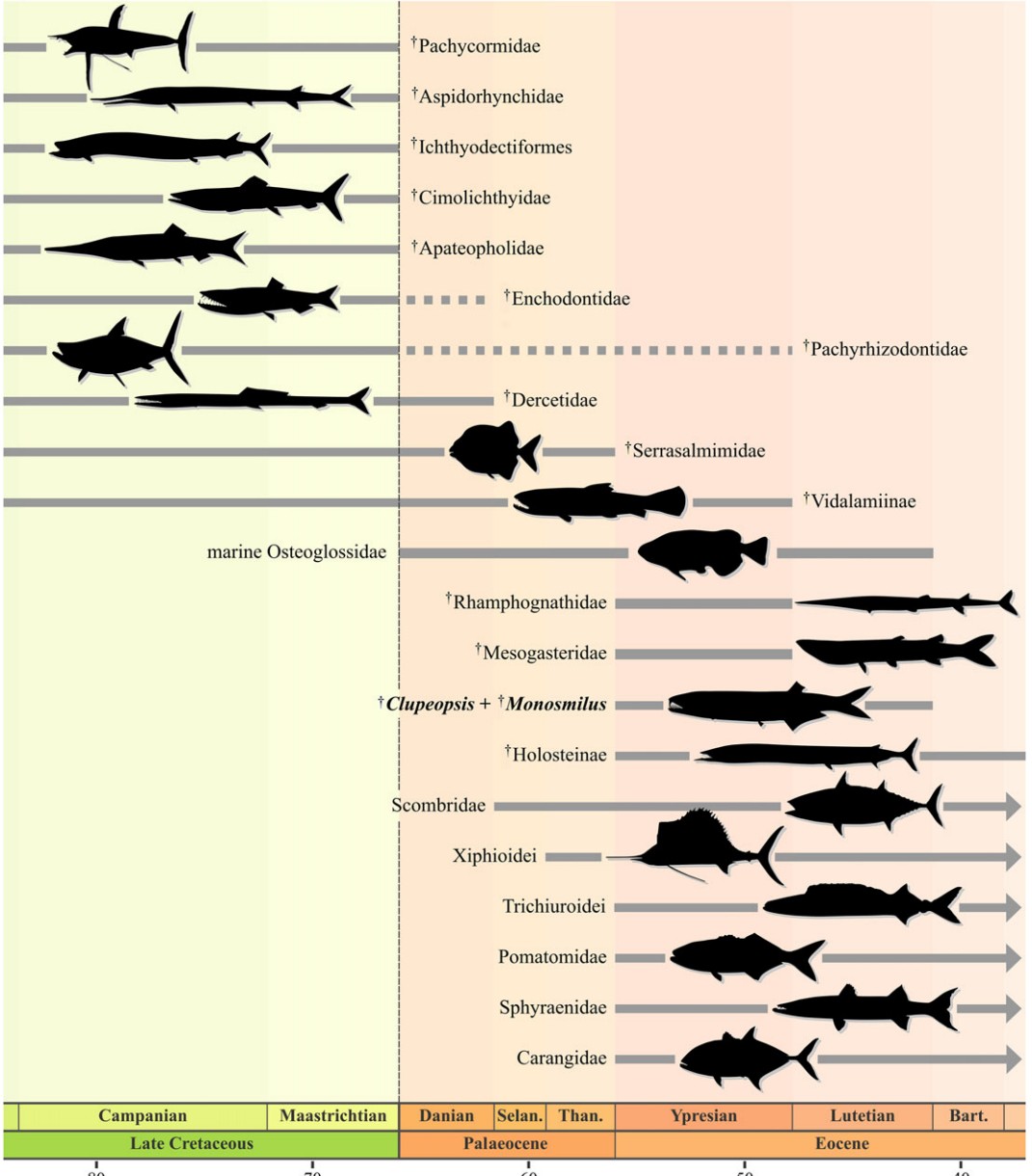

**Figure 4.** Medium-to-large-bodied predatory actinopterygians in shallow marine environments through time, showing faunal turnover across the K/Pg extinction and the early Palaeogene. †Aspidorhynchids are reported from continental deposits in the Palaeocene [59], but their last marine occurrences are Maastrichtian [60]. †Enchodontids are reported from Danian marine deposits [61], but this is possibly due to reworking from Maastrichtian layers [9]. The last occurrence of †Holosteinae extends into the early Oligocene (Rupelian; not shown here). An arrow on the right end of the illustrated range indicates survivorship to the present day. Ranges are approximated to stage level and derived from [14,15,62–70]. Silhouettes not to scale, and adapted from [8,14,15,62,65,66,71–74] and images from Triebold Paleontology, Inc. (http://trieboldpaleontology.com/). Abbreviations: Bart., Bartonian; Selan., Selandian; Than., Thanetian.

The Early Cretaceous †*Cynoclupea* joins *Chirocentrus* as another putative relative of the engraulid clade, leading Malabarba and Di Dario to hypothesize that planktivorous engraulids are derived from a piscivorous ancestor [50]. This dietary transition had not been inferred elsewhere in clupeoids [48], but has clear precedents in other groups of modern (e.g. *Polyodon* [55], scombrine scombrids [56]) and extinct (e.g. edentulous †pachycormiforms [57]) fishes. Our interpretation of †*Monosmilus* and †*Clupeopsis* could provide additional support for this inferred trajectory of trophic evolution. Future analyses including morphological data and expanded molecular datasets will represent a vital test of the relationships among modern lineages, hopefully providing a more robust framework for considering the meaning of †*Clupeopsis* and †*Monosmilus* for the evolution of feeding strategies within clupeiforms.

Independent of their remarkable anatomical specializations, †Clupeopsis and †Monosmilus provide a rare and unexpected perspective on the fossil record of total group engraulids. Despite including over 150 extant species that are widely distributed, engraulids have a very poor fossil record both in absolute terms and in comparison to other clupeoids [21,58]. Until the discovery of the early Eocene (Ypresian) crown engraulid †Eoengraulis from Bolca, definitive fossil anchovies were known only from Miocene or younger deposits [19]. Dated at roughly 5 million years older than †Eoengraulis, †Clupeopsis is the oldest representative of the engraulid total group, providing a fossil-based minimum age for the divergence between this radiation and its yet unresolved sister lineage. However, the divergence between engraulids and other clupeiforms is likely to substantially predate the age of †Clupeopsis, with molecular clock estimates placing the origin of the engraulid total group anywhere from the Early Cretaceous to the earliest Palaeogene [48,52,54]. The persistence of fanged stem engraulids for an interval of at least 8 million years from the early Ypresian to the Lutetian points to the early–middle Eocene as an interval of high engraulid morphological and ecological disparity, with these trophically specialized, early-diverging lineages coexisting with highly nested members of the crown radiation that, like most modern species, were probably planktivorous [19].

## 4.5. Trophic and environmental diversification among non-acanthomorph marine teleosts in the Palaeogene

One of the most striking features of turnover among marine fishes associated with the K/Pg boundary is the extinction (e.g. †pachycormids, †ichthyodectiforms, †cimolichthyids and some other aulopiforms) or decimation and marginalization (e.g. †pachyrhizodontids, †dercetids) of many of the dominant groups of large, predatory Cretaceous fishes [9–11], followed by the Palaeogene origins of several independent lineages of predatory acanthomorphs (e.g. sphyraenids, carangids, xiphioids, scombrids, trichiuroids, along with the short-lived †rhamphognathids and †mesogasterids) [8,56] associated with the broader radiation of spiny-rayed fishes (figure 4) [3,7].

In comparison, the patterns of trophic diversification and environmental shifts among non-acanthomorph lineages around the K/Pg have received less attention. However, there are striking cases of apparently short-lived predatory non-acanthomorph lineages in shallow marine settings that, like the origin of modern groups of predatory acanthomorphs, might represent a response to new ecological opportunities in the early Palaeogene. The most conspicuous example is a radiation of osteoglossids, a group that in the modern day is restricted to freshwater settings. Marine osteoglossids range in age from early Palaeocene (Danian) to middle Eocene (Lutetian) and include several named genera and unnamed forms [14,75]. Marine osteoglossids partially overlapped temporally, and in some cases co-occurred with, an early Eocene (Ypresian) to early Oligocene (Rupelian) shallow-water lineage of large-bodied (ca 1 m) paralepidid aulopiforms (barracudinas), a group that in the modern day is associated with meso- or bathypelagic settings [15]. These new Palaeogene radiations were joined by a handful of late-surviving examples of predatory marine neopterygian lineages that arose in the Mesozoic, all of which (e.g. †serrasalmimid pycnodontiforms [62], †vidalamine amiids [63] and possibly †pachyrhizodontids [64]) appear to have gone extinct by the end of the Eocene. The raptorial stem engraulids †Clupeopsis and †Monosmilus add to this diverse assortment of early Palaeogene non-acanthomorph predators in shallow-water settings (figure 4). It is unclear whether the ecology characterizing these genera evolved in the Cretaceous (as hypothesized by Malabarba and Di Dario [50]), or if it only arose in the Palaeogene. However, it is apparent that Palaeogene predatory fish guilds in shallow marine settings were composed of phylogenetically diverse members, paralleling the pattern of numerous now-extinct predatory mammal and archosaur lineages (e.g. †creodonts, †mesonychids, †sparassodonts and terrestrial crocodilians [76,77]) that coexisted with modern radiations (e.g. carnivorans) in terrestrial ecosystems at the same time.

Data accessibility. New tomograms generated for this study, as well as surface models for segmented anatomical structures, are archived at the Dryad Digital Repository (https://doi.org/10.5061/dryad.f4qrfj6ss).

Authors' contributions. M.F. and A.C. conceived the idea. P.D.G. led the team that collected the holotype of †Monosmilus and established stratigraphic context for it. H.T.B. and A.C. segmented μCT data. E.S. established stratigraphic context for †Clupeopsis and wrote geological portions of the supplement. A.C. produced the figures, and A.C. and M.F. drafted the manuscript. All authors contributed to revision of the manuscript.

Competing interests. We declare we have no competing interests.

Funding. This work was supported by funding from the Department of Earth and Environmental Sciences of the University of Michigan (Scott Turner Student Research Grant Award 2017, to A.C.) and by the Society of

Systematic Biologists (2017 SSB Graduate Student Research Award, to A.C.). Field research in Pakistan was sponsored by the Geological Survey of Pakistan and by the Smithsonian Institution (U.S. PL-480 grant no. SFC-70760400). This study includes data produced in the CTEES facility at University of Michigan, supported by the Department of Earth & Environmental Sciences and College of Literature, Science, and the Arts.

Acknowledgements. We thank D. Nelson and R. Singer (UMMZ) for access to modern comparative materials; A. Folie (RBINS) for arranging a loan of †*Clupeopsis*, which was transported by R. Close (University of Birmingham); S. Giles (University of Birmingham) for guidance on Blender; and Z. Randall (UF) for permission to download data stored on Morphosource. Devin Bloom (Western Michigan University), Juan Marcos Mirande (Unidad Ejecutora Lillo-CONICET) and Oksana Vernygora (University of Alberta) provided thoughtful and constructive reviews that improved this contribution.

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
