## [Reviewer comments · Royal Society Open Science]

Review History

RSOS-192260.R0 (Original submission)

Review form: Reviewer 1 (Devin D. Bloom)

Is the manuscript scientifically sound in its present form?

Yes

Are the interpretations and conclusions justified by the results?

No

Is the language acceptable?

Yes

Do you have any ethical concerns with this paper?

No

Have you any concerns about statistical analyses in this paper?

No

Recommendation?

Major revision is needed (please make suggestions in comments)

Comments to the Author(s)

Capobianco and co-authors present a study describing two new fossil anchovies. This is an interesting and important study because there are very few fossils for this ecologically and economically important group of fishes, which limits our ability to interpret the evolutionary history of the clade. Intriguingly, the fossils described by the authors suggest morphological novelty in the form of large fang-like teeth which implies ecological diversity not previously known from the fossil record. Overall the manuscript is well composed and the description establishing these species as engraulids seems robust.

I like this paper a lot and I'm excited about the implications. However, I think there are several areas that I outline below.

The diagnoses of these two new taxa could be improved with some quantitative characters, or by using more definitive characters. Terms like "massive" (Example: Pg 5 line 34 and again Pg 8 line 14) are not helpful in a diagnosis. How big is "massive"? It is commonplace in extant descriptions to quantify characters. For example, if the fang is an easy character identify and measure then give as a proportion of jaw or head length. This allows future descriptions to clearly distinguish specimens as either the same or different species. This applies throughout the diagnoses. Another example: What is a "long maxilla? Does it look like *Thryssa setirostris* (Longjaw anchovy), which extends approximately half the body length, or something else? Descriptions of fossils are not my area of expertise, but I would encourage you to follow extant descriptions that quantify these diagnostic characters and remove ambiguous terms.

It is a little unclear if the two specimens described consist entirely of the skulls or if the skulls were the focus of the description. For example, on Pg 5 line 13 the authors state "an incomplete individual preserved...with an almost complete skull". This leaves me wondering if there is a skull with more body fragments or the skull is the entire specimen recovered. This makes it unclear if the length reported for both specimens is just the skull or the entire specimen. I don't have access to the supplementary materials, but I am left wondering if figures 1 and 2 show the entire fossil? I think just some simple text clarifying this would clean this up easily.

The ambiguity with the size and completeness of the fossils leads to further questions about the interpretation of these fossils. For example, figure 4 seems to imply the authors estimated the body size of these fossils and suggests these fossil engraulids were as large as carangids and billfishes! In section e of the discussion the fossil engraulids are discussed as medium to large bodied fishes, but there is little incorporation with what is known about extant Clupeiformes. Body size has been studied across all Clupeiformes (Bloom et al., 2020, 2018)(note the latter study just came out, so I am not faulting the authors for not including it in their discussion). Given that the authors are discussing the ecological diversification of anchovies, and attempting to put these fossils in the same category as other large predators it seems relevant incorporate what is known about extant engraulids (and other Clupeiformes) and the ecology exhibited by this group.

So just how large are these fossil engraulids? I used the size reported (278mm and 104mm), assumed these were head lengths (?), and estimated the standard length using a regression of head length and standard length from a dataset that includes ~900 specimens of extant Clupeiformes. Based on these data the fossils would measure in at 1236mm and 470mm. Obviously some assumptions are required (but $R^2 = 0.72$, $P = 2.2e-16$), but this at least gives a pretty good idea of the size of these fishes. For context, the largest extant clupeiform is *Chirocentrus*, which reaches about 1 meter standard length, so roughly the same size as *Monosmilus*. The authors are welcome to use these estimates if they are helpful.

Describing trophic diversity of clupeiformes during the early Paleogene as being primarily small-bodied suspension feeders is problematic. Most contemporary clupeiformes are planktivores

(numerically) but not suspension feeders and I don't know of any evidence to suggest the situation was different 40-60ma. I sense the authors are using suspension feeding and filter feeding interchangeably, but they are not synonymous, and assuming that all clupeiform planktivores are filter feeders. Many planktivorous engraulids (e.g. *Engraulis mordax*) feed using a variety of strategies, including something much better described alternating between biting and filter feeding. Raptorial feeding/biting behavior in zooplanktivores is typically not considered to be suspension/filter feeding even though it isn't necessary entirely selective. Thus, I think it is important that the authors clearly define their terminology and revise this part of the discussion to avoid confusion (see Gibson, 1988; James, 1987; Sanderson and Wassersug, 1993). I don't think that this study provides any evidence that suspension feeders evolved from piscivores. Zooplanktivory almost certainly evolved before suspension feeding in Clupeiformes. Additionally, the placement of these two piscivorous fossil species as stem engraulids, along with the diet ancestral state reconstructions in Egan et al (2018), can also be explained by these fossil lineages independently having evolved piscivory from a zooplanktivorous (non-suspension feeder) ancestor.

Finally, and this is not a small point, I found it very strange that there was no discussion of the implications for the age and timing of diversification of engraulids. There are numerous time-calibrated studies of Clupeiforms (Bloom and Lovejoy, 2014; Egan et al., 2018; e.g. Lavoué et al., 2013), which was reviewed in detail by Bloom and Egan (Bloom and Egan, 2018). There are two paragraphs dedicated to interpreting the ecology of these Paleogene fossils, which requires a fair bit of speculation, yet no discussion of the age of engraulids, a topic that the authors have direct evidence to address! I strongly encourage adding this to the discussion.

Minor comments:

The introduction includes a number of common names without associated Latin names

Pg 2 Second paragraph of introduction you mention *Clupeopsis* specifically but not *Monosmilus*. This creates a little confusion. Is *Clupeopsis* previously described? A little background here would help.

Pg 12 line 8: I'm not sure what "total-group engraulids" is referring to exactly. Does this mean all of *Engraulidae* or all of the fossil engraulids? Or something else?

In section d of the discussion I would cite Whitehead (Whitehead et al., 1988) or some other reference of your choice for the claim that "all Clupeiforms with caniniform teeth are piscivorous".

I want to end by re-emphasizing that I really like this study and think it has a lot to offer. I also have never suggested more than a single paper I authored be incorporated in a review. In this case, there are really only 2-3 people working on the topic, so it was inevitable (though I feel uncomfortable about it).

Don't hesitate to reach out if I can be of any help.

Signed,
Devin Bloom

References

Bloom, D.D., Burns, M.D., Schriever, T.A., 2018. Evolution of body size and trophic position in migratory fishes: a phylogenetic comparative analysis of Clupeiformes (anchovies, herring, shad and allies). *Biol. J. Linn. Soc.* 125, 302–314.

- Bloom, D.D., Egan, J.P., 2018. Systematics of Clupeiformes and testing for ecological limits on species richness in a trans-marine / freshwater clade. *Neotrop. Ichthyol.* 16, 1–14. doi:10.1590/1982-0224-20180095
- Bloom, D.D., Kolmann, M., Foster, K., Watrous, H., 2020. Mode of miniaturisation influences body shape evolution in New World anchovies (Engraulidae). *J. Fish Biol.* 96, 194–201. doi:10.1111/jfb.14205
- Bloom, D.D., Lovejoy, N.R., 2014. The evolutionary origins of diadromy inferred from a time-calibrated phylogeny for Clupeiformes (herring and allies). *Proc. R. Soc. B Biol. Sci.* 281, 20132081. doi:10.1098/rspb.2013.2081
- Egan, J.P., Bloom, D.D., Kuo, C.-H., Hammer, M.P., Tongnunui, P., Iglésias, S.P., Sheaves, M., Grudpan, C., Simons, A.M., 2018. Phylogenetic analysis of trophic niche evolution reveals a latitudinal herbivory gradient in Clupeoidei (herrings, anchovies, and allies). *Mol. Phylogenet. Evol.* 124, 151–161. doi:10.1016/j.ympev.2018.03.011
- Gibson, R.N., 1988. Development, morphometry and particle retention capability of the gill rakers in the herring, *Clupea harengus* L. *J. Fish Biol.* 32, 949–962. doi:10.1111/j.1095-8649.1988.tb05438.x
- James, A.G., 1987. Feeding Ecology, Diet and Field-Based Studies on Feeding Selectivity of the Cape Anchovy *Engraulis-Capensis* Gilchrist. *South African J. Mar. Sci.* 5, 673–692.
- Lavoué, S., Miya, M., Musikasinthorn, P., Chen, W.-J., Nishida, M., 2013. Mitogenomic Evidence for an Indo-West Pacific Origin of the Clupeoidei (Teleostei: Clupeiformes). *PLoS One* 8, e56485. doi:10.1371/journal.pone.0056485
- Sanderson, S.L., Wassersug, R., 1993. Convergent and alternative designs for vertebrate suspension feeding, in: Hanken, J., Hall, B.K. (Eds.), *The Skull. Volume 3. Functional and Evolutionary Mechanisms.* University of Chicago Press, Chicago, IL, pp. 37–112.
- Whitehead, P.J.P., Nelson, G.J., Wongratana, T., 1988. *FAO species catalogue, vol. 7. Clupeoid fishes of the world (Suborder Clupeoidei).* UNDP FAO, Rome.

Review form: Reviewer 2 (Marcos Mirande)

Is the manuscript scientifically sound in its present form?

Yes

Are the interpretations and conclusions justified by the results?

Yes

Is the language acceptable?

Yes

Do you have any ethical concerns with this paper?

No

Have you any concerns about statistical analyses in this paper?

No

Recommendation?

Accept with minor revision (please list in comments)

Comments to the Author(s)

Review of "Large-bodied sabre-toothed anchovies reveal unanticipated ecological diversity in early Palaeogene teleosts"

The manuscript provides mainly the description of a new species assigned to a new genus related with *Clupeopsis* and inferences about the diversification of these fishes in the early Palaeogene. The description is well-written and results are sounding and interesting. Authors made a very good use of modern techniques for fossil inspection, which were essential to discover many of the details provided.

The single concern I had while reading the manuscript was why not to describe the new species in *Clupeopsis*? Morphological divergence itself doesn't justify the creation of a new genus if the species form a clade, but I understand that the authors aren't sure that both species really form a monophyletic group. Maybe some sentence could be added regarding the decision of erecting a new genus.

I liked the doubt casted about the relationships of the clade (instead of concluding something from ambiguous evidence). Also, I liked both genera treated as incertae sedis in *Clupeoidei* instead of defining a new suprageneric taxon.

Lastly, I don't know if it will add something relevant for the paper, but just for completion of the examined literature, I add up a(n) (auto-)reference below.

Sincerely, Marcos Mirande

page,line.

2,13. Maybe "was the extinction" is better than "is the extinction"
 10,40. Well, although self-referencial, the authors are omitting Mirande(2016:
<https://onlinelibrary.wiley.com/doi/abs/10.1111/cla.12171>;
https://morphobank.org/index.php/Projects/ProjectDocuments/project_id/2435) where I
 analyzed 106 species of *Clupeiformes* and 44 loci in the context of an Actinopterygian phylogeny.
 Is far to be "genome-wide" but is the most comprehensive analysis until the moment.

Review form: Reviewer 3

Is the manuscript scientifically sound in its present form?

Yes

Are the interpretations and conclusions justified by the results?

Yes

Is the language acceptable?

Yes

Do you have any ethical concerns with this paper?

No

Have you any concerns about statistical analyses in this paper?

No

Recommendation?

Accept with minor revision (please list in comments)

Comments to the Author(s)

This is a very interesting and valuable contribution by Capobianco and co-authors describing a new fossil lineage of large piscivorous clupeomorphs. This study contributes to our knowledge of the ecological diversity and trait evolution within the group. The newly obtained ct-data provides anatomical information otherwise not available for examination and description.

Overall, the authors did a wonderful job providing description and figures of the new and previously reported fossil material. I recommend this paper for publication with a few minor revisions and comments included in the attached pdf copy of the manuscript (Appendix A).

Most importantly, make sure that information in the abstract is consistent with the morphological description (palatal vs vomerine fangs), label supplementary figures (S3 a, b; S9 a, b), in the 'systematic paleontology' section for the new taxon (*Monosmilus*) please add a paragraph justifying its placement within the Clupeiformes.

Thank you for the very interesting study.

Decision letter (RSOS-192260.R0)

04-Mar-2020

Dear Mr Capobianco

On behalf of the Editors, I am pleased to inform you that your Manuscript RSOS-192260 entitled "Large-bodied sabre-toothed anchovies reveal unanticipated ecological diversity in early Palaeogene teleosts" has been accepted for publication in Royal Society Open Science subject to minor revision in accordance with the referee suggestions. Please find the referees' comments at the end of this email.

The reviewers and handling editors have recommended publication, but also suggest some minor revisions to your manuscript. Therefore, I invite you to respond to the comments and revise your manuscript.

- Ethics statement

- Data accessibility

It is a condition of publication that all supporting data are made available either as supplementary information or preferably in a suitable permanent repository. The data accessibility section should state where the article's supporting data can be accessed. This section should also include details, where possible of where to access other relevant research materials such as statistical tools, protocols, software etc can be accessed. If the data has been deposited in an external repository this section should list the database, accession number and link to the DOI for all data from the article that has been made publicly available. Data sets that have been

deposited in an external repository and have a DOI should also be appropriately cited in the manuscript and included in the reference list.

If you wish to submit your supporting data or code to Dryad (<http://datadryad.org/>), or modify your current submission to dryad, please use the following link:
<http://datadryad.org/submit?journalID=RSOS&manu=RSOS-192260>

- **Competing interests**

- **Authors' contributions**

- **Acknowledgements**

- **Funding statement**

Because the schedule for publication is very tight, it is a condition of publication that you submit the revised version of your manuscript before 13-Mar-2020. Please note that the revision deadline will expire at 00.00am on this date. If you do not think you will be able to meet this date please let me know immediately.

When submitting your revised manuscript, you will be able to respond to the comments made by the referees and upload a file "Response to Referees" in "Section 6 - File Upload". You can use this to document any changes you make to the original manuscript. In order to expedite the

processing of the revised manuscript, please be as specific as possible in your response to the referees. We strongly recommend uploading two versions of your revised manuscript:

If your manuscript is newly submitted and subsequently accepted for publication, you will be asked to pay the article processing charge, unless you request a waiver and this is approved by Royal Society Publishing. You can find out more about the charges at <https://royalsocietypublishing.org/rsos/charges>. Should you have any queries, please contact openscience@royalsociety.org.

on behalf of Dr Julia Brenda Desojo (Associate Editor) and Kevin Padian (Subject Editor)
 openscience@royalsociety.org

Subject Editor Comments to Author (Professor Kevin Padian):

Comments to the Author:

Thanks for your submission. The referees are all quite positive about the contribution but each has concerns ranging from the precision of the diagnostic criteria to the question whether you could have put your animal in Clupeopsis. Please address each of these points when you submit your revision. Best wishes.

Reviewer comments to Author:

Reviewer: 1

Comments to the Author(s)

Capobianco and co-authors present a study describing two new fossil anchovies. This is an interesting and important study because there are very few fossils for this ecologically and economically important group of fishes, which limits our ability to interpret the evolutionary history of the clade. Intriguingly, the fossils described by the authors suggest morphological novelty in the form of large fang-like teeth which implies ecological diversity not previously known from the fossil record. Overall the manuscript is well composed and the description establishing these species as engraulids seems robust.

I like this paper a lot and I'm excited about the implications. However, I think there are several areas that I outline below.

The diagnoses of these two new taxa could be improved with some quantitative characters, or by using more definitive characters. Terms like "massive" (Example: Pg 5 line 34 and again Pg 8 line 14) are not helpful in a diagnosis. How big is "massive"? It is commonplace in extant descriptions to quantify characters. For example, if the fang is an easy character identify and measure then give as a proportion of jaw or head length. This allows future descriptions to clearly distinguish specimens as either the same or different species. This applies throughout the diagnoses. Another example: What is a "long maxilla? Does it look like *Thryssa setirostris* (Longjaw anchovy), which extends approximately half the body length, or something else? Descriptions of fossils are not my area of expertise, but I would encourage you to follow extant descriptions that quantify these diagnostic characters and remove ambiguous terms.

It is a little unclear if the two specimens described consist entirely of the skulls or if the skulls were the focus of the description. For example, on Pg 5 line 13 the authors state "an incomplete individual preserved....with an almost complete skull". This leaves me wondering if there is a skull with more body fragments or the skull is the entire specimen recovered. This makes it unclear if the length reported for both specimens is just the skull or the entire specimen. I don't have access to the supplementary materials, but I am left wondering if figures 1 and 2 show the entire fossil? I think just some simple text clarifying this would clean this up easily.

The ambiguity with the size and completeness of the fossils leads to further questions about the interpretation of these fossils. For example, figure 4 seems to imply the authors estimated the body size of these fossils and suggests these fossil engraulids were as large as carangids and billfishes! In section e of the discussion the fossil engraulids are discussed as medium to large bodied fishes, but there is little incorporation with what is known about extant Clupeiformes. Body size has been studied across all Clupeiformes (Bloom et al., 2020, 2018)(note the latter study just came out, so I am not faulting the authors for not including it in their discussion). Given that the authors are discussing the ecological diversification of anchovies, and attempting to put these

fossils in the same category as other large predators it seems relevant incorporate what is known about extant engraulids (and other Clupeiformes) and the ecology exhibited by this group.

So just how large are these fossil engraulids? I used the size reported (278mm and 104mm), assumed these were head lengths (?), and estimated the standard length using a regression of head length and standard length from a dataset that includes ~900 specimens of extant Clupeiformes. Based on these data the fossils would measure in at 1236mm and 470mm. Obviously some assumptions are required (but $R^2 = 0.72$, $P = 2.2e-16$), but this at least gives a pretty good idea of the size of these fishes. For context, the largest extant clupeiform is *Chirocentrus*, which reaches about 1 meter standard length, so roughly the same size as *Monosmilus*. The authors are welcome to use these estimates if they are helpful.

Describing trophic diversity of clupeiformes during the early Paleogene as being primarily small-bodied suspension feeders is problematic. Most contemporary clupeiformes are planktivores (numerically) but not suspension feeders and I don't know of any evidence to suggest the situation was different 40-60ma. I sense the authors are using suspension feeding and filter feeding interchangeably, but they are not synonymous, and assuming that all clupeiform planktivores are filter feeders. Many planktivorous engraulids (e.g. *Engraulis mordax*) feed using a variety of strategies, including something much better described alternating between biting and filter feeding. Raptorial feeding/biting behavior in zooplanktivores is typically not considered to be suspension/filter feeding even though it isn't necessary entirely selective. Thus, I think it is important that the authors clearly define their terminology and revise this part of the discussion to avoid confusion (see Gibson, 1988; James, 1987; Sanderson and Wassersug, 1993). I don't think that this study provides any evidence that suspension feeders evolved from piscivores. Zooplanktivory almost certainly evolved before suspension feeding in Clupeiformes. Additionally, the placement of these two piscivorous fossil species as stem engraulids, along with the diet ancestral state reconstructions in Egan et al (2018), can also be explained by these fossil lineages independently having evolved piscivory from a zooplanktivorous (non-suspension feeder) ancestor.

Finally, and this is not a small point, I found it very strange that there was no discussion of the implications for the age and timing of diversification of engraulids. There are numerous time-calibrated studies of Clupeiforms (Bloom and Lovejoy, 2014; Egan et al., 2018; e.g. Lavoué et al., 2013), which was reviewed in detail by Bloom and Egan (Bloom and Egan, 2018). There are two paragraphs dedicated to interpreting the ecology of these Paleogene fossils, which requires a fair bit of speculation, yet no discussion of the age of engraulids, a topic that the authors have direct evidence to address! I strongly encourage adding this to the discussion.

Minor comments:

The introduction includes a number of common names without associated Latin names

Pg 2 Second paragraph of introduction you mention *Clupeopsis* specifically but not *Monosmilus*. This creates a little confusion. Is *Clupeopsis* previously described? A little background here would help.

Pg 12 line 8: I'm not sure what "total-group engraulids" is referring to exactly. Does this mean all of *Engraulidae* or all of the fossil engraulids? Or something else?

In section d of the discussion I would cite Whitehead (Whitehead et al., 1988) or some other reference of your choice for the claim that "all Clupeiforms with caniniform teeth are piscivorous".

I want to end by re-emphasizing that I really like this study and think it has a lot to offer. I also have never suggested more than a single paper I authored be incorporated in a review. In this

case, there are really only 2-3 people working on the topic, so it was inevitable (though I feel uncomfortable about it).

Don't hesitate to reach out if I can be of any help.

Signed,
Devin Bloom

References

Bloom, D.D., Burns, M.D., Schriever, T.A., 2018. Evolution of body size and trophic position in migratory fishes: a phylogenetic comparative analysis of Clupeiformes (anchovies, herring, shad and allies). *Biol. J. Linn. Soc.* 125, 302–314.

Bloom, D.D., Egan, J.P., 2018. Systematics of Clupeiformes and testing for ecological limits on species richness in a trans-marine / freshwater clade. *Neotrop. Ichthyol.* 16, 1–14.
doi:10.1590/1982-0224-20180095

Bloom, D.D., Kolmann, M., Foster, K., Watrous, H., 2020. Mode of miniaturisation influences body shape evolution in New World anchovies (Engraulidae). *J. Fish Biol.* 96, 194–201.
doi:10.1111/jfb.14205

Bloom, D.D., Lovejoy, N.R., 2014. The evolutionary origins of diadromy inferred from a time-calibrated phylogeny for Clupeiformes (herring and allies). *Proc. R. Soc. B Biol. Sci.* 281, 20132081.
doi:10.1098/rspb.2013.2081

Egan, J.P., Bloom, D.D., Kuo, C.-H., Hammer, M.P., Tongnunu, P., Iglésias, S.P., Sheaves, M., Grudpan, C., Simons, A.M., 2018. Phylogenetic analysis of trophic niche evolution reveals a latitudinal herbivory gradient in Clupeoidei (herrings, anchovies, and allies). *Mol. Phylogenet. Evol.* 124, 151–161. doi:10.1016/j.ympev.2018.03.011

Gibson, R.N., 1988. Development, morphometry and particle retention capability of the gill rakers in the herring, *Clupea harengus* L. *J. Fish Biol.* 32, 949–962. doi:10.1111/j.1095-8649.1988.tb05438.x

James, A.G., 1987. Feeding Ecology, Diet and Field-Based Studies on Feeding Selectivity of the Cape Anchovy *Engraulis-Capensis* Gilchrist. *South African J. Mar. Sci.* 5, 673–692.

Lavoué, S., Miya, M., Musikasinthorn, P., Chen, W.-J., Nishida, M., 2013. Mitogenomic Evidence for an Indo-West Pacific Origin of the Clupeoidei (Teleostei: Clupeiformes). *PLoS One* 8, e56485.
doi:10.1371/journal.pone.0056485

Sanderson, S.L., Wassersug, R., 1993. Convergent and alternative designs for vertebrate suspension feeding, in: Hanken, J., Hall, B.K. (Eds.), *The Skull. Volume 3. Functional and Evolutionary Mechanisms*. University of Chicago Press, Chicago, IL, pp. 37–112.

Whitehead, P.J.P., Nelson, G.J., Wongratana, T., 1988. *FAO species catalogue, vol. 7. Clupeoid fishes of the world (Suborder Clupeoidei)*. UNDP FAO, Rome.

Reviewer: 2

Comments to the Author(s)

Review of "Large-bodied sabre-toothed anchovies reveal unanticipated ecological diversity in early Palaeogene teleosts"

The manuscript provides mainly the description of a new species assigned to a new genus related

with *Clupeopsis* and inferences about the diversification of these fishes in the early Palaeogene. The description is well-written and results are sounding and interesting. Authors made a very good use of modern techniques for fossil inspection, which were essential to discover many of the details provided.

The single concern I had while reading the manuscript was why not to describe the new species in *Clupeopsis*? Morphological divergence itself doesn't justify the creation of a new genus if the species form a clade, but I understand that the authors aren't sure that both species really form a monophyletic group. Maybe some sentence could be added regarding the decision of erecting a new genus.

I liked the doubt casted about the relationships of the clade (instead of concluding something from ambiguous evidence). Also, I liked both genera treated as *incertae sedis* in *Clupeioidi* instead of defining a new suprageneric taxon.

Lastly, I don't know if it will add something relevant for the paper, but just for completion of the examined literature, I add up a(n) (auto-)reference below.

Sincerely, Marcos Mirande

page,line.

2,13. Maybe "was the extinction" is better than "is the extinction"
 10,40. Well, although self-referencial, the authors are omitting Mirande(2016:
<https://onlinelibrary.wiley.com/doi/abs/10.1111/cla.12171>;
https://morphobank.org/index.php/Projects/ProjectDocuments/project_id/2435) where I
 analyzed 106 species of *Clupeiformes* and 44 loci in the context of an *Actinopterygian* phylogeny.
 Is far to be "genome-wide" but is the most comprehensive analysis until the moment.

Reviewer: 3

Comments to the Author(s)

This is a very interesting and valuable contribution by Capobianco and co-authors describing a new fossil lineage of large piscivorous *clupeomorphs*. This study contributes to our knowledge of the ecological diversity and trait evolution within the group. The newly obtained ct-data provides anatomical information otherwise not available for examination and description.

Overall, the authors did a wonderful job providing description and figures of the new and previously reported fossil material. I recommend this paper for publication with a few minor revisions and comments included in the attached pdf copy of the manuscript.

Most importantly, make sure that information in the abstract is consistent with the morphological description (palatal vs vomerine fangs), label supplementary figures (S3 a, b; S9 a, b), in the 'systematic paleontology' section for the new taxon (*Monosmilus*) please add a paragraph justifying its placement within the *Clupeiformes*.

Thank you for the very interesting study.

Author's Response to Decision Letter for (RSOS-192260.R0)

See Appendix B.

Decision letter (RSOS-192260.R1)

09-Apr-2020

Dear Mr Capobianco,

It is a pleasure to accept your manuscript entitled "Large-bodied sabre-toothed anchovies reveal unanticipated ecological diversity in early Palaeogene teleosts" in its current form for publication in Royal Society Open Science.

on behalf of Dr Julia Brenda Desojo (Associate Editor) and Kevin Padian (Subject Editor)
openscience@royalsociety.org

Appendix A**ROYAL SOCIETY
OPEN SCIENCE****Large-bodied sabre-toothed anchovies reveal unanticipated ecological diversity in early Palaeogene teleosts**

Journal:	Royal Society Open Science
Manuscript ID	RSOS-192260
Article Type:	Research
Date Submitted by the Author:	31-Dec-2019
Complete List of Authors:	Capobianco, Alessio; University of Michigan, Department of Earth and Environmental Sciences; University of Michigan, Museum of Paleontology Beckett, Hermione; University of Oxford, Department of Earth Sciences; King's High School for Girls Steurbaut, Etienne; Royal Belgian Institute of Natural Sciences; KU Leuven Gingerich, Phillip; University of Michigan, Department of Earth and Environmental Sciences; University of Michigan, Museum of Paleontology Carnevale, Giorgio; Università di Torino, Dipartimento di Scienze della Terra Friedman, Matthew; University of Michigan, Department of Earth and Environmental Sciences; University of Michigan, Museum of Paleontology
Subject:	palaeontology < BIOLOGY, evolution < BIOLOGY
Keywords:	Clupeomorpha, Palaeogene, computed tomography, ecological release, ichthyology, piscivory
Subject Category:	Organismal and Evolutionary Biology

Author-supplied statements

Relevant information will appear here if provided.

Ethics

Does your article include research that required ethical approval or permits?:

This article does not present research with ethical considerations

Statement (if applicable):

CUST_IF_YES_ETHICS :No data available.

Data

It is a condition of publication that data, code and materials supporting your paper are made publicly available. Does your paper present new data?:

Yes

Statement (if applicable):

New tomograms generated for this study, as well as surface models for segmented anatomical structures, will be archived in a Dryad repository <link to be populated upon acceptance>.

Surface models for segmented anatomical structures described and figured in the paper are available for reviewers and Editors to download following this link:

<https://drive.google.com/open?id=1p7f90rLjkryEfwRbrvhhaCEcWOL9Ymej>

Conflict of interest

I/We declare we have no competing interests

Statement (if applicable):

CUST_STATE_CONFLICT :No data available.

Authors' contributions

This paper has multiple authors and our individual contributions were as below

Statement (if applicable):

M.F. and A.C. conceived the idea. H.T.B. and A.C. segmented $\hat{1}\frac{1}{4}$ CT data. E.S. established stratigraphic context for *Clupeopsis* and wrote geological portions of the supplement. A.C. produced the figures.

A.C. and M.F. drafted the manuscript. All authors contributed to revision of the manuscript.

Large-bodied sabre-toothed anchovies reveal unanticipated ecological diversity in early Palaeogene teleosts

short title: Sabre-toothed Eocene anchovies

Alessio Capobianco^{1,2*}, Hermione T. Beckett^{3,4}, Etienne Steurbaut^{5,6}, Philip D. Gingerich^{1,2}, Giorgio Carnevale⁷, Matt Friedman^{1,2}

¹Department of Earth and Environmental Sciences, University of Michigan, Ann Arbor, MI, USA

²Museum of Paleontology, University of Michigan, Ann Arbor, MI, USA

³Department of Earth Sciences, University of Oxford, Oxford, UK

⁴King's High School for Girls, Warwick, UK

⁵Royal Belgian Institute of Natural Sciences, Brussels, Belgium

⁶KU Leuven, Leuven, Belgium

⁷Dipartimento di Scienze della Terra, Università degli Studi di Torino, Torino, Italy

Abstract

Many modern groups of marine fishes first appear in the fossil record during the early Palaeogene (66–40 million years ago), including iconic predatory lineages of spiny-rayed fishes that appear to have originated in response to ecological roles left empty after the Cretaceous/Palaeogene extinction. The hypothesis of extinction-mediated ecological release likewise predicts that other fish groups have adopted novel predatory ecologies. Here we report remarkable trophic innovation in early Palaeogene clupeiforms (herrings and allies), a group whose modern representatives are generally small-bodied suspension feeders. Two forms, the early Eocene (Ypresian) †*Clupeopsis* from Belgium and a new genus from the middle Eocene (Lutetian) of Pakistan, bear conspicuous features indicative of predatory ecology, including large size, long gapes, and caniniform dentition. Most remarkable is the presence of a single, massive palatal fang¹ offset from the midline in both. Numerous features place these taxa on the engraulid (anchovy) stem as the earliest known representatives of the clade. The identification of large-bodied, piscivorous anchovies contributes to an emerging picture of a phylogenetically diverse guild of predatory ray-finned fishes in early Palaeogene marine settings, which include completely extinct lineages alongside members of modern marine groups and taxa that are today restricted to freshwater or deep-sea environments.

Keywords: Clupeomorpha, computed tomography, ecological release, ichthyology, Palaeogene, piscivory

1. Introduction

Body fossils [1-3], otoliths [4], ichthyoliths [5, 6] and molecular clocks [7, 8] point to the early Palaeogene as a time of remarkable diversification and innovation among marine fishes. Most of the groups familiar from modern marine ecosystems appeared during this interval, along with their distinctive morphological adaptations. So striking is this pattern that the evolution of marine teleosts after the Eocene has been described by some as “mere tinkering” [1]. One of the clearest impacts of the Cretaceous/Palaeogene (K/Pg) extinction on teleosts is the extinction of large predatory taxa [9-11], which has been implicated in permitting the subsequent diversification of lineages like mackerels, barracudas, billfishes, and jacks that first appear near the Paleocene–Eocene boundary [3, 8, 12, 13]. All of these examples belong to a group of fishes called acanthomorphs, or spiny-rayed teleosts, which represent the dominant fish group in marine settings since the beginning of the Cenozoic [1, 2]. However, the hypothesis that opportunity arising from the K/Pg extinctions fueled diversification predicts that other groups may have also experimented with new roles in the early Palaeogene, although this has been little investigated (but see [14, 15]) despite significance for understanding the structure of marine faunas at that time.

Here we report a new early–middle Eocene clade of large-bodied clupeiform (herrings and anchovies) fishes, the anatomy of which has been revealed by micro-computed tomography (μ CT). This group, represented by †*Clupeopsis straeleni* from the Ypresian of Belgium and a new genus and species from the Lutetian of Pakistan, is characterized by remarkable dental specializations: a single row of enlarged dentary teeth combined with a single massive vomerine tang that extends to the ventral margin of the mandibular symphysis. These fossils force a reconsideration of trophic diversity among marine clupeiforms in the early Palaeogene, which are otherwise represented by small-bodied, probable suspension feeders apparently similar to the vast majority of living clupeiforms [16-19]. More broadly, they point to previously unappreciated trophic innovation in an early Palaeogene marine setting that has not persisted to the modern day.

2. Materials and methods

(a) Micro-computed tomography (μ CT) scanning

The holotypes of †*Clupeopsis straeleni* (MRHNB IG 8630) and GSP-UM 37, as well as representative examples of extant clupeiforms, were imaged using Nikon XT H 225ST industrial μ CT scanners at the University of Michigan and the Natural History Museum, London. Scan parameters are provided in electronic supplementary material. Reconstructed datasets were visualized and segmented using Mimics v. 19.0 (Materialise, Belgium). Models of segmented skeletal elements were exported as surface files (.ply)

and rendered as high-quality images in Blender v. 2.79 (blender.org). Individual scanning parameters of new tomograms generated for this study are provided in electronic supplementary material.

(b) Comparative material

μ CT scans obtained from the following formalin-fixed specimens of extant clupeiform species were examined as comparative material.

Clupeidae. *Clupea harengus* UF 184063 (Morphosource media M44470), *Odaxothrissa mento* UMMZ 195016.

Pristigasteridae. *Ilisha elongata* UF 143661 (Morphosource media M44747), *Odontognathus mucronatus* UF 135948 (Morphosource media M44474).

Chirocentridae. *Chirocentrus dorab* UMMZ 238306, *Chirocentrus nudus* UMMZ 213502.

Engraulidae. *Lycengraulis grossidens* UMMZ 143053, *Setipinna [Lycothrissa] crocodilus* UMMZ 209911.

In addition to the material listed here, further observations of clupeiform osteology were derived from descriptive accounts [20-25]. Following best practices in the accessibility of tomographic data [26], we have made tomograms, .mcs files, and .ply files of segmented structures available as supplementary files in Data Dryad.

(c) Institutional abbreviations

GSP-UM, Geological Survey of Pakistan, Quetta, Pakistan, specimens collected during joint expeditions with the University of Michigan Museum of Paleontology; MRHNB, Musée Royal d'Histoire Naturelle de Belgique (now RBINS), Brussels, Belgium; RBINS, Royal Belgian Institute of Natural Sciences, Brussels, Belgium; UF, University of Florida, Gainesville, USA; UMMZ, University of Michigan Museum of Zoology, Ann Arbor, USA.

(d) Dagger symbol

Following the convention of ref. [27], the dagger symbol (“†”) precedes extinct groups.

3. Results

Systematic palaeontology.

Teleostei Müller, 1845 (~~ref. [28]~~)

Clupeomorpha Greenwood, Rosen, Weitzman and Myers, 1966 (~~ref. [29]~~)

Clupeiformes **Bleeker, 1859**  *sensu* Grande, 1985 (~~ref. [20]~~)

Clupeoidei Jordan, 1923 (~~ref. [30]~~) *sensu* Greenwood, Rosen, Weitzman and Myers, 1966 (~~ref. [29]~~)

Engrauloidea Grande, 1985 (~~ref. [20]~~)

†*Clupeopsis straeleni* Casier, 1946 (~~ref. [31]~~)

**(a) Material**

MRHNB IG 8630 (holotype n. 276), Royal Belgian Institute of Natural Sciences, Brussels, Belgium.
Holotype and only known specimen, representing an incomplete individual preserved three-dimensionally
with an almost complete skull (figure 1; Electronic Supplementary Material [ESM] figures S1-S6);
specimen 278 mm in length.

**(b) Locality and horizon**

Dubois clay pit, Chièvres, Hainaut, Belgium, 3.7 m above Ypresian basal gravel [32]. This represents the
basal part of the Orchies Clay Member, providing an age constraint of approximately 54.40–54.05 million
23 years ago (Ma). The Orchies Clay Member is interpreted as being deposited in an outer neritic marine
setting, but its base appears to represent a shallower facies [33]. Additional details are provided in electronic
supplementary material.

**(c) Diagnosis**

Clupeiform with triangular-shaped skull in lateral and dorsal views; vomer bearing two large tooth pits;
single, massive vomerine fang, extending ventrally beyond the mandibular symphysis; dorsolaterally
oriented pre-epiotic fossa; long, robust and slightly curved toothless maxilla; straight and robust dentary

[revised manuscript text omitted]

(d) Implications for clupeiform evolution and ecological diversification

A combination of well-developed caniniform dentition, slender mandibles consistent with rapid jaw closing, and relatively large size suggests a predatory—and likely piscivorous—feeding ecology for †*Clupeopsis* and †*Monosmilus* [46]. The majority of modern clupeiforms are suspension feeders [47]. Similar ecologies are inferred for most fossil forms [16-18, 48] (but see the Early Cretaceous †*Cynoclupea* [49]). However, among extant clupeoids, piscivory is the second-most common dietary strategy, and appears to have evolved multiple times independently from zooplanktivory [47]. Extant piscivorous taxa have a variety of tooth morphologies, including complete absence of teeth, but all clupeiforms with caniniform teeth are piscivorous. These include members of Chirocentridae (*Chirocentrus*), Clupeidae (*Odaxothrissa*), Pristigasteridae (*Chirocentrodon bleakerianus*) and Engraulidae (*Setipinna* [*Lycothrissa*] *crocodilus* and *Lycengraulis*). Among these, only chirocentrids show development of the caniniform dentition comparable to that of †*Monosmilus* and †*Clupeopsis*.

The relationships among major clupeoid lineages is unclear. Genomic-scale data are available for few species [50], while taxonomically well-sampled molecular phylogenies include only a handful of loci [47]. The placement of chirocentrids is especially unclear, with this group resolved in a variety of positions by different molecular analyses [47, 51]. Di Dario [21] hypothesized a sister-group relationship between engraulids and chirocentrids on the basis of shared features in multiple anatomical systems. Combined with subsequent phylogenetic interpretation of the Early Cretaceous †*Cynoclupea* as sister to the putative chirocentrid/engraulid clade, this led to the hypothesis that suspension feeding 
[revised manuscript text omitted]

- [4] Schwarzhan, W., Schulz-Mirbach, T., Lombarte, A. & Tuset, V.M. 2017 The origination and rise of teleost otolith diversity during the Mesozoic. *Research & Knowledge* **3**, 5-8.
- [5] Sibert, E.C., Friedman, M., Hull, P., Hunt, G. & Norris, R.D. 2018 Two pulses of morphological diversification in Pacific pelagic fishes following the Cretaceous-Palaeogene mass extinction. *Proceedings of the Royal Society B* **285**, 20181194.
- [6] Sibert, E.C. & Norris, R.D. 2015 New Age of Fishes initiated by the Cretaceous-Paleogene mass extinction. *Proceedings of the National Academy of Sciences of the USA* **112**, 8537-8542.
- [7] Alfaro, M.E., Faircloth, B.C., Harrington, R.C., Sorenson, L., Friedman, M., Thacker, C.E., Oliveros, C.H., Černý, D. & Near, T.J. 2018 Explosive diversification of marine fishes at the Cretaceous-Paleogene boundary. *Nature Ecology & Evolution* **2**, 688-696.
- [8] Harrington, R.C., Faircloth, B.C., Eytan, R.I., Smith, W.L., Near, T.J., Alfaro, M.E. & Friedman, M. 2016 Phylogenomic analysis of carangimorph fishes reveals flatfish asymmetry arose in a blink of the evolutionary eye. *BMC Evolutionary Biology* **16**, 224.
- [9] Cavin, L. 2002 Effects of the Cretaceous-Tertiary boundary event on fishes. In *Geological and Biological Effects of Impact Events* (eds. E. Buffetaut & C. Koeberl), pp. 141-158. Berlin, Springer.
- [10] Cavin, L. & Martin, C. 1995 Les Actinoptérygiens et la limite Crétacé-Tertiaire. *Geobios MS* **28**, 183-188.
- [11] Friedman, M. 2009 Ecomorphological selectivity among marine teleost fishes during the end-Cretaceous extinction. *Proceedings of the National Academy of Sciences of the USA* **106**, 5218-5223.
- [12] Miya, M., Friedman, M., Satoh, T.O., Takeshima, H., Sado, T., Iwasaki, W., Yamanoue, Y., Nakatani, M., Mabuchi, K., Inoue, J.G., et al. 2013 Evolutionary origin of the Scombridae (tunas and mackerels): members of a Paleogene adaptive radiation with 14 other pelagic fish families. *PLoS ONE* **8**, e73535.
- [13] Santini, F., Carnevale, G. & Sorenson, L. 2014 First timetree of Sphraenidae (Percomorpha) reveals a middle Eocene crown age and an Oligo-Miocene radiation. *Italian Journal of Zoology* **82**, 133-142.
- [14] Capobianco, A., Foreman, E. & Friedman, M. In press A Paleocene (Danian) marine osteoglossid (Teleostei: Osteoglossomorpha) from the Nuussuaq Basin of Greenland, with a brief review of Palaeogene marine bonytongue fishes. *Papers in Palaeontology*.
- [15] Marramà, G. & Carnevale, G. 2017 Morphology, relationships and palaeobiology of the Eocene barracudina †*Holosteus esocinus* (Aulopiformes: Paralepididae) from Monte Bolca, Italy *Zoological Journal of the Linnean Society* **181**, 209-228.
- [16] Marramà, G. & Carnevale, G. 2014 Eocene round herring from Monte Bolca, Italy. *Acta Palaeontologica Polonica* **60**, 701-710.
- [17] Marramà, G. & Carnevale, G. 2015 The Eocene sardine †*Bolcaichthys catopygopterus* (Woodward, 1901) from Monte Bolca, Italy: osteology, taxonomy, and paleobiology. *Journal of Vertebrate Paleontology* **35**, e101490.
- [18] Marramà, G. & Carnevale, G. 2015 An Eocene anchovy from Monte Bolca, Italy: The earliest known record for the family Engraulidae. *Geological Magazine* **153**, 84-94.
- [19] Marramà, G. & Carnevale, G. 2018 *Eoalosa janvieri* gen. et sp. nov., a new clupeid fish (Teleostei, Clupeiformes) from the Eocene of Monte Bolca, Italy. *Paläontologische Zeitschrift* **92**, 107-120.

[20] Grande, L. 1985 Recent and fossil clupeomorph fishes with materials for revision of the subgroups of
clupeoids. *Bulletin of the American Museum of Natural History* **181**, 231-372.
- [21] Di Dario, F. 2009 Chirocentrids as engrauloids: evidence from suspensorium, branchial arches, and
infraorbital bones (Clupeomorpha, Teleostei). *Zoological Journal of the Linnean Society* **156**, 363-383.
- [22] Nelson, G.J. 1970 The hyobranchial apparatus of teleostean fishes of the families Engraulidae and
Chirocentridae. *American Museum Novitates* **2410**, 1-30.
- [23] Nelson, G.J. 1967 Gill arches of some teleostean fishes of the family Clupeidae. *Copeia* **1967**, 389-399.
- [24] Bardack, D. 1965 Anatomy and evolution of chirocentrid fishes. *The University of Kansas*
*Paleontological Contributions, Vertebrata* **10**, 1-86.
- [25] Ridewood, W.G. 1904 On the cranial osteology of the clupeoid fishes. *Proceedings of the Zoological*
*Society of London* **1904**, 448-493.
- [26] Davies, T.G., Rahman, I.A., Lautenschlager, S., Cunningham, J.A., Asher, R.J., Barrett, P.M., Bates, K.T.,
Bengston, S., Benson, R.B.J., Boyer, D.M., et al. 2017 Open data and digital morphology. *Proceedings of*
*the Royal Society B* **284**.
- [27] Patterson, C. & Rosen, D.E. 1977 A review of ichthyodectiform and other Mesozoic teleost fishes and
the theory and practice of classifying fossil. *Bulletin of the American Museum of Natural History* **158**, 81-
172.
- [28] Müller, J. 1845 Über den Bau und die Grenzen der Ganoiden, und über das natürliche System der
Fische. *Physikalisch-Mathematische Abhandlungen der königlichen Akademie der Wissenschaften zu*
*Berlin* **1845**, 117-216.
- [29] Greenwood, P.H., Rosen, D.E., Weitzman, S.H. & Myers, G.S. 1966 Phyletic studies of the teleostean
fishes, with a provisional classification of living forms. *Bulletin of the American Museum of Natural History*
**131**, 339-456.
- [30] Jordan, D.S. 1923 A classification of fishes including families and genera as far as known. *Stanford*
*University Publications, University Series, Biological Sciences* **3**, 77-243.
- [31] Casier, E. 1946 La faune ichthyologique de l'Yprésien de la Belgique. *Mémoires du Musée Royal*
*d'Histoire Naturelle de Belgique* **104**.
- [32] Delvaux, E. & Ortlieb, J. 1887 Les poissons fossiles de l'argile yprésienne de Belgique. – Description
paléontologique accompagnée de documents stratigraphiques pour servir à l'étude monographique de
cet étage. *Annales de la Société Géologique du Nord* **15**, 50-66.
- [33] King, C., Gale, A.S. & Barry, T.L. 2016 *A revised correlation of Tertiary rocks in the British Isles and*
*adjacent areas of NW Europe*. London, Geological Society of London; 719 p.
- [34] Gingerich, P.D., Russell, D.E., Sigogneau-Russell, D., Hartenberger, J.-L., Shah, S.M.I., Hassan, M., Rose,
36 K.D. & Ardrey, R.H. 1979 Reconnaissance survey and vertebrate paleontology of some Paleocene and
37 Eocene formations in Pakistan. *Contributions from the Museum of Paleontology, University of Michigan*
**25**, 105-116.
- [35] Gingerich, P.D., Arif, M. & Clyde, W.C. 1995 New archaeocetes (Mammalia, Cetacea) from the middle
Eocene Domanda Formation of the Sulaiman Range, Punjab (Pakistan). *Contributions from the Museum*
*of Paleontology, University of Michigan* **29**, 291-330.
- [36] Raju, S.N. 1974 Three new species of the genus *Monognathus* and the leptocephali of the order
Saccopharyngiformes. *Fishery Bulletin* **72**, 547-562.
- [37] Bertelsen, E. & Nielsen, J. 1987 The deepsea eel family Monognathidae (Pisces Anguilliformes).
*Steenstrupia* **13**, 141-198.
- [38] Smith, J.L.B. 1955 The genus *Pyramodon* Smith & Radcliffe 1913. *Annals and Magazine of Natural*
*History* (**12**) **8**, 545-550.
- [39] Robins, C.H. & Robins, C.R. 1975 New genera and species of dysommene and synphobranchine eels
(Synphobranchidae) with an analysis of the Dysommeneae. *Proceedings of the Academy of Natural*
*Sciences of Philadelphia* **127**, 249-280.

[40] McCosker, J.E. 1977 The osteology, classification, and relationships of the eel family Ophichthidae.
*Proceedings of the California Academy of Sciences* **41**, 1-123.
- [41] Goody, P.C. 1969 The relationships of certain Upper Cretaceous teleosts with special reference to the
myctophoids. *Bulletin of the British Museum (Natural History) Geology*, 1-255.
- [42] Starks, E.C. 1926 Bones of the ethmoid region of the fish skull. *Stanford University Publications,*
*University Series, Biological Sciences* **4**, 137-338.
- [43] Le Danois, Y. 1974 Étude ostéo-myologique et revision systématique de la famille des Lophiidae,
(pédiculates haploptérygiens). *Mémoires du Muséum national d'Histoire naturelle, Nouvelle Série, Série*
*A, Zoologique* **91**, 1-127.
- [44] Böhlke, E.B., Böhlke, J.E., Leiby, M.M., McCosker, J.E., Bertelsen, E., Robins, C.H., Robins, C.R., Smith,
D.G., Tighe, K.A., Nielsen, J.G., et al. 1989 *Orders Anguilliformes and Saccopharyngiformes*. New Haven,
Sears Foundation for Marine Research, Yale University.
- [45] Gosline, W.A. 1951 The osteology and classification of the ophichthid eels of the Hawaiian Islands.
*Pacific Science* **5**, 298-320.
- [46] Mihalitsis, M. & Bellwood, D. 2019 Functional implications of dentition-based morphotypes in
piscivorous fishes. *Royal Society Open Science* **6**, 190040.
- [47] Egan, J.P., Bloom, D.D., Kuo, C.-H., Hammer, M.P., Tongnunui, P., Iglesias, S.P., Sheaves, M., Grudpan,
C. & Simons, A.W. 2018 Phylogenetic analysis of trophic niche evolution reveals a latitudinal herbivory
gradient in Clupeoidei (herrings, anchovies, and allies). *Molecular Biology and Evolution* **124**, 151-161.
- [48] Grande, L. 1984 Paleontology of the Green River Formation, with a review of the fish fauna. *Geological*
*Survey of Wyoming, Bulletin* **63**, 1-333.
- [49] Malabarba, M.C. & di Dario, F. 2017 A new predatory herring-like fish (Teleostei: Clupeiformes) from
the Early Cretaceous of Brazil, and implications for relationships in the Clupeoidei. *Zoological Journal of*
*the Linnean Society* **180**, 175-194.
- [50] Hughes, L.C., Ortí, G., Huang, Y., Sun, Y., Baldwin, C.C., Thompson, A.W., Arcila, D., Betancur-R., R., Li,
C., Becker, L., et al. 2018 Comprehensive phylogeny of ray-finned fishes (Actinopterygii) based on
transcriptomic and genomic data. *Proceedings of the National Academy of Sciences of the USA* **115**, 6249-
6254.
- [51] Lavoué, S., Miya, M., Musiakasinthorn, P., Chen, W.-J. & Nishida, M. 2013 Mitogenomic evidence for
an Indo-West Pacific origin of the Clupeoidei (Teleostei: Clupeiformes). *PLoS ONE* **8**, e56485.
- [52] Grande, L. & Bemis, W.E. 1991 Osteology and phylogenetic relationships of fossil and recent
paddlefishes (Polyodontidae) with comments on interrelationships of Acipenseriformes. *Society of*
*Vertebrate Paleontology Memoir* **1**, 1-121.
- [53] Friedman, M., Feilich, K.L., Beckett, H.T., Alfaro, M.E., Faircloth, B.C., Cerny, D., Miya, M., Near, T.J. &
Harrington, R.C. 2019 A phylogenomic framework for pelagiarian fishes (Acanthomorpha: Percomorpha)
highlights mosaic radiation in the open ocean. *Proceedings of the Royal Society B* **286**, rspb.2019.1502.
- [54] Friedman, M. 2012 Parallel evolutionary trajectories underlie the origin of giant suspension-feeding
whales and bony fishes. *Proceedings of the Royal Society B* **279**, 944-951.
- [55] Grande, L. & Nelson, G.J. 1985 Interrelationships of fossil and recent achovies (Teleostei:
Engrauloidea) and description of a new species from the Miocene of Cyprus. *American Museum Novitates*
**2826**, 1-16.
- [56] Hilton, E.J. & Lavoué, S. 2018 A review of the systematic biology of fossil and living bony-tongue fishes,
Osteoglossomorpha (Actinopterygii: Teleostei). *Neotropical Ichthyology* **16**, e180031.
- [57] Vullo, R., Cavin, L., Khalloufi, B., Amaghaz, M., Bardet, N., Jalil, N.-E., Khaldoune, F. & Gheerbrant, E.
2017 A unique Cretaceous-Paleogene lineage of pirhana-jawed pycnodont fishes. *Scientific Reports* **7**,
6802.
- [58] O'Leary, M.A., Bouaré, M.L., Claeson, K.M., Heilbronn, K., Hill, R.V., McCartney, J., Sessa, J.A., Sissoko,
F., Tapinala, L., Wheeler, E., et al. 2019 Stratigraphy and paleobiology of the Upper Cretaceous-lower

Paleogene sediments from the trans-Saharan seaway in Mali. *Bulletin of the American Museum of Natural History* **436**, 1-177.

[59] Taverne, L. 1980 Ostéologie et position systématique du genre *Platinx* (Pisces, Teleostei) de l'éocène du Monte Bolca (Italie). *Academie Royale de Belgique, Bulletin de la Classe des Sciences, 5, série* **66**, 873-889.

[60] Gunnell, G.F., Morgan, M.E., Maas, M.C. & Gingerich, P.D. 1995 Comparative paleoecology of Paleogene and Neogene mammalian faunas: Trophic structure and composition. *Palaeogeography, Palaeoclimatology, Palaeoecology* **115**, 265-286.

[61] Prevosti, F.J., Forasiepi, A. & Zimicz, N. 2013 The evolution of the Cenozoic terrestrial mammalian predator guild in South America: competition or replacement? *Journal of Mammalian Evolution* **20**, 3-21.

[62] Bryant, L.J. 1987 *Belonostomus* (Teleostei: Aspidorhynchidae) from the late Paleocene of North Dakota. *PaleoBios* **43**, 1-3.

[63] Friedman, M. 2012 Ray-finned fishes (Osteichthyes, Actinopterygii) from the type Maastrichtian, the Netherlands and Belgium. In *Fossils of the type Maastrichtian (Part 1). Scripta Geologica Special Issue 8* (eds. J.W.M. Jagt, S.K. Donovan & E.A. Jagt-Yazykova), pp. 112-142. Leiden.

[64] Grande, L. & Bemis, W.E. 1998 A comprehensive phylogenetic study of amiid fishes (Amiidae) based on comparative skeletal anatomy. An empirical search for interconnected patterns of natural history. *Journal of Vertebrate Paleontology* **18**, 1-696.

[65] Bannikov, A.F. 2008 Revision of the atheriniform fish genera *Rhamphognathus* Agassiz and *Mesogaster* Agassiz (Teleostei) from the Eocene of Bolca, northern Italy. *Studi e Ricerche sui Giacimenti Terziari di Bolca* **9**, 65-76.

[66] Carnevale, G., Bannikov, A.F., Marramà, G., Tyler, J.C. & Zorzin, R. 2014 The Pesciara-Monte Postale *Fossil-Lagerstätte*: 2. Fishes and other vertebrates. *Rendiconti dell Società Paleontologica Italiana* **4**, 37-63.

[67] Guinot, G. & Cavin, L. 2016 'Fish' (Actinopterygii and Elasmobranchii) diversification patterns through deep time. *Biological Reviews* **91**, 950-981.

[68] Solé, F., Noiret, C., Desmares, D., Adnet, S., Taverne, L., De Putter, T., Mees, F., Yans, J., Steeman, T., Louwye, S., et al. 2019 Reassessment of historical sections from the Paleogene marine margin of the Congo Basin reveals an almost complete absence of Danian deposits. *Geoscience Frontiers* **10**, 1039–1063.

[69] Adolfssen, J.S., Milan, J. & Friedman, M. 2017 Review of the Danian vertebrate fauna of southern Scandinavia. *Bulletin of the Geological Survey of Denmark* **65**, 1-23.

[revised manuscript text omitted]

Abbreviations: Bart., Bartonian; Selan., Selandian; Than., Thanetian.

182x212mm (300 x 300 DPI)

Appendix B

Alessio Capobianco, PhD Candidate

Ann Arbor, 27 March 2020

To the Editorial Board,

We wish to thank you and the three referees for carefully reviewing our manuscript and providing helpful feedback that has strengthened this contribution. In the sections below, we present point-by-point responses to the comments of referees, noting the changes that we have made in the manuscript. Our comments are indicated in **bold**.

All changes from the original manuscript are highlighted in the document with tracked changes submitted alongside the 'clean' version.

Please do not hesitate to contact me if you require any further information about this revision.

Alessio Capobianco

Response to Referee 1

The diagnoses of these two new taxa could be improved with some quantitative characters, or by using more definitive characters. Terms like “massive” (Example: Pg 5 line 34 and again Pg 8 line 14) are not helpful in a diagnosis. How big is “massive”? It is commonplace in extant descriptions to quantify characters. For example, if the fang is an easy character identify and measure then give as a proportion of jaw or head length. This allows future descriptions to clearly distinguish specimens as either the same or different species. This applies throughout the diagnoses. Another example: What is a “long maxilla? Does it look like *Thryssa setirostris* (Longjaw anchovy), which extends approximately half the body length, or something else? Descriptions of fossils are not my area of expertise, but I would encourage you to follow extant descriptions that quantify these diagnostic characters and remove ambiguous terms.

The diagnosis of †*Clupeopsis* was changed to include more quantitative characters. These include “single vomerine fang, extending ventrally beyond the mandibular symphysis when jaws closed and representing the largest tooth”; “robust and slightly curved toothless maxilla measuring approximately 75% of neurocranium length”; “largest dentary tooth approximately 20% of length of orbital cavity”. It is more difficult to define quantitative characters as proportions of other body parts in †*Monosmilus*, due to the fragmentariness of the only known specimen. We included “largest dentary tooth approximately 70% of length of orbital cavity” as a quantitative proportional character in the †*Monosmilus* diagnosis.

It is a little unclear if the two specimens described consist entirely of the skulls or if the skulls were the focus of the description. For example, on Pg 5 line 13 the authors state “an incomplete individual preserved....with an almost complete skull”. This leaves me wondering if there is a skull with more body fragments or the skull is the entire specimen recovered. This makes it unclear if the length reported for both specimens is just the skull or the entire specimen. I don’t have access to the supplementary materials, but I am left wondering if figures 1 and 2 show the entire fossil? I think just some simple text clarifying this would clean this up easily.

We changed both Material sub-sections to hopefully provide a clearer description of what is preserved in the two specimens and what the reported lengths are referring to.

The Material sub-section for †*Clupeopsis* now reads: “MRHNB IG 8630 (holotype n. 276), Royal Belgian Institute of Natural Sciences, Brussels, Belgium. Holotype and only known specimen, representing a three-dimensionally preserved individual comprising an almost complete skull plus incomplete body extending to the level of the dorsal fin (figure 1; Electronic Supplementary Material [ESM] figures S1-S6). The specimen measures 278 mm from the tip of the snout to the broken posterior end of the body.”.

The Material sub-section for †*Monosmilus* now reads: “GSP-UM 37, Geological Survey of Pakistan, Quetta, Pakistan. Holotype and only known specimen, an incomplete but three-dimensionally preserved skull broken anteriorly at the tip of the snout and posteriorly in advance of the occipital condyle (figure 2; ESM figures S5, S7-10). The specimen measures 104 mm between these broken surfaces. Collected during a November-December 1977 field season of the Geological Survey of Pakistan and the University of Michigan Museum of Paleontology [35].”

The ambiguity with the size and completeness of the fossils leads to further questions about the interpretation of these fossils. For example, figure 4 seems to imply the authors estimated the

body size of these fossils and suggests these fossil engraulids were as large as carangids and billfishes! In section e of the discussion the fossil engraulids are discussed as medium to large bodied fishes, but there is little incorporation with what is known about extant Clupeiforms. Body size has been studied across all Clupeiformes (Bloom et al., 2020, 2018)(note the latter study just came out, so I am not faulting the authors for not including it in their discussion). Given that the authors are discussing the ecological diversification of anchovies, and attempting to put these fossils in the same category as other large predators it seems relevant incorporate what is known about extant engraulids (and other Clupeiforms) and the ecology exhibited by this group. So just how large are these fossil engraulids? I used the size reported (278mm and 104mm), assumed these were head lengths (?), and estimated the standard length using a regression of head length and standard length from a dataset that includes ~900 specimens of extant Clupeiformes. Based on these data the fossils would measure in at 1236mm and 470mm. Obviously some assumptions are required (but $R^2 = 0.72$, $P = 2.2e-16$), but this at least gives a pretty good idea of the size of these fishes. For context, the largest extant clupeiform is *Chirocentrus*, which reaches about 1 meter standard length, so roughly the same size as *Monosmilus*. The authors are welcome to use these estimates if they are helpful.

The silhouettes in Fig. 4 are not to scale, they are just meant to represent the taxonomic turnover across the K/Pg extinction and during the early Palaeogene. We added “Silhouettes not to scale” in Fig. 4 caption.

The lengths used by the referee to derive body length estimates for the two fossil taxa are not head lengths, as explained in the comment above (this has now been clarified in the manuscript). Because the referee’s dataset of extant Clupeiformes including head lengths and standard lengths has not been published yet, we measured head lengths and standard lengths in chirocentrid and engraulid specimen photographs taken from FishBase (<https://www.fishbase.us/>). We used these data to estimate approximate standard lengths of †*Clupeopsis* and †*Monosmilus*. These were then compared with the distribution of standard lengths in extant clupeiforms, using data from Bloom *et al.*, 2018. Methods and results are detailed in Supplementary Figure S13. The following lines were included at the beginning of section d) of the Discussion to clarify what previously were ambiguous statements about the size of fossil sabre-toothed anchovies: “†*Clupeopsis* and †*Monosmilus* are large in comparison to extant anchovies and most clupeiforms more generally. The incompleteness of the two specimens precludes exact measurements, but estimates can be made based on proportions in living species. Linear regressions of body length in respect to head length in extant clupeiforms yield body lengths of just below half meter and one meter for †*Clupeopsis* and †*Monosmilus* respectively (ESM figure S13).”

Describing trophic diversity of clupeiformes during the early Paleogene as being primarily small-bodied suspension feeders is problematic. Most contemporary clupeiformes are planktivores (numerically) but not suspension feeders and I don't know of any evidence to suggest the situation was different 40-60ma. I sense the authors are using suspension feeding and filter feeding interchangeably, but they are not synonymous, and assuming that all clupeiform planktivores are filter feeders. Many planktivorous engraulids (e.g. *Engraulis mordax*) feed using a variety of strategies, including something much better described alternating between biting and filter feeding. Raptorial feeding/biting behavior in zooplanktivores is typically not considered to be suspension/filter feeding even though it isn't necessary entirely selective. Thus, I think it is important that the authors clearly define their terminology and revise this part of the discussion

to avoid confusion (see Gibson, 1988; James, 1987; Sanderson and Wassersug, 1993). I don't think that this study provides any evidence that suspension feeders evolved from piscivores. Zooplanktivory almost certainly evolved before suspension feeding in Clupeiformes. Additionally, the placement of these two piscivorous fossil species as stem engraulids, along with the diet ancestral state reconstructions in Egan et al (2018), can also be explained by these fossil lineages independently having evolved piscivory from a zooplanktivorous (non-suspension feeder) ancestor.

We recognize that we misused the term 'suspension feeding' in the manuscript. As we also agree with the referee's comments about the ancestral nature of planktivory (and specifically zooplanktivory) within clupeiforms, we substituted 'suspension feeding' with 'planktivory' throughout the manuscript.

Finally, and this is not a small point, I found it very strange that there was no discussion of the implications for the age and timing of diversification of engraulids. There are numerous time-calibrated studies of Clupeiforms (Bloom and Lovejoy, 2014; Egan et al., 2018; e.g. Lavoué et al., 2013), which was reviewed in detail by Bloom and Egan (Bloom and Egan, 2018). There are two paragraphs dedicated to interpreting the ecology of these Paleogene fossils, which requires a fair bit of speculation, yet no discussion of the age of engraulids, a topic that the authors have direct evidence to address! I strongly encourage adding this to the discussion.

We briefly discussed this point in section d) of the Discussion ("Dated at roughly 5 million years older than †*Eoengraulis*, †*Clupeopsis* is the oldest representative of the engraulid total group, providing a fossil-based minimum age for the divergence between this radiation and its yet unresolved sister lineage."). We believe that the use of †*Clupeopsis* as a calibration point will not drastically change estimates for the time of origin of total-group engraulids, as published estimates are already much older than the age of †*Clupeopsis*. We added the following sentence to clarify this point: "However, the divergence between engraulids and other clupeiforms is likely to substantially predate the age of †*Clupeopsis*, with molecular clock estimates placing the origin of the engraulid total group anywhere from the Early Cretaceous earliest Paleogene [48, 52, 54]."

The introduction includes a number of common names without associated Latin names.

Latin names added to the common names between parentheses in the Introduction.

Pg 2 Second paragraph of introduction you mention *Clupeopsis* specifically but not *Monosmilus*. This creates a little confusion. Is *Clupeopsis* previously described? A little background here would help.

Sentence changed to "[...] represented by the previously described †*Clupeopsis straeleni* [...] and a new genus and species [...]" for clarification.

Pg 12 line 8: I'm not sure what "total-group engraulids" is referring to exactly. Does this mean all of Engraulidae or all of the fossil engraulids? Or something else?

'Total-group' is a commonly used terminology in palaeontological literature to indicate crown group + stem group. So, in this case, it indicates all extant engraulids, plus all extinct engraulids nested within crown Engraulidae, plus all stem engraulids (which are, by definition, all extinct).

In section d of the discussion I would cite Whitehead (Whitehead et al., 1988) or some other reference of your choice for the claim that “all Clupeiforms with caniniform teeth are piscivorous”.
Reference added as suggested.

Response to Referee 2

General comments. The single concern I had while reading the manuscript was why not to describe the new species in *Clupeopsis*? Morphological divergence itself doesn't justify the creation of a new genus if the species form a clade, but I understand that the authors aren't sure that both species really form a monophyletic group. Maybe some sentence could be added regarding the decision of erecting a new genus.

We recognize that the decision of erecting a new genus for a new species closely related to †*Clupeopsis straeleni* has some degree of subjectivity. However, we believe that the morphological differences between †*C. straeleni* and †*M. chureloides* are enough to warrant the erection of a new genus for the latter. We added a Notes sub-section after the Diagnosis of †*M. chureloides* to clarify our choice and highlight the differences between the two taxa.

Page 2, line 13. Maybe "was the extinction" is better than "is the extinction".

Change made as suggested.

Page 10, line 40. Well, although self-referential, the authors are omitting Mirande (2016: <https://onlinelibrary.wiley.com/doi/abs/10.1111/cla.12171>; https://morphobank.org/index.php/Projects/ProjectDocuments/project_id/2435) where I analyzed 106 species of Clupeiformes and 44 loci in the context of an Actinopterygian phylogeny. Is far to be "genome-wide" but is the most comprehensive analysis until the moment.

Reference added to the list of published taxonomically well-sampled phylogenies of Clupeiformes.

Response to Referee 3

General comments. Most importantly, make sure that information in the abstract is consistent with the morphological description (palatal vs vomerine fangs), label supplementary figures (S3 a, b; S9 a, b), in the 'systematic paleontology' section for the new taxon (*Monosmilus*) please add a paragraph justifying its placement within the Clupeiformes.

We fixed the abstract to make it consistent with the morphological description and labeled supplementary figures S3 and S9. We added a couple of lines in the Discussion to clarify the placement of †*Monosmilus* within Clupeiformes (see below our response to the comment on page 8, line 14). However, we do not think it is necessary to add these justifications to the Systematic Palaeontology section as well.

Page 2, line 43. Make sure this is consistent with the rest of the manuscript where you mention vomerine fangs and not palatal fangs.

Changed “palatal” with “vomerine” to be consistent.

Page 2, line 44.

Added “of the neurocranium, suspensorium and branchial skeleton” to specify the skeletal regions that display engraulid-like features.

Page 3, line 33. In the abstract, it's referred to as a palatal fang...please clarify if it's palatal or vomerine.

See above the change made to the abstract for consistency.

Page 4, line 54.

Reference added for Bleeker, 1859.

Page 6, line 11.

Added “along the midline” to clarify how the parietals are separated by the frontals and by the supraoccipital.

Page 6, line 28. It would be helpful for the readers if Fig. S3 (particularly parts (a) and (b)) was labeled similar to Fig. S2 indicating this and other features mentioned in the description.

Figures S3 and S9 were modified to add labels indicating bones and features mentioned in the description.

Page 7, line 21. Are there any traces of the infraorbital sensory canal?

The infraorbital sensory canal can be clearly seen in the anterior portion of the greatly expanded posterior infraorbital. Traces of an infraorbital sensory canal can be also seen in the elongated anterior infraorbital. We added “, based on their position and canal-bearing” to this line.

Page 8, line 14. Because this is the first description of the new taxon, I would expect to see a justification for why it was identified as a clupeomorph clupeiform.

As explained in the Discussion (section 4.b), the attribution of †*Monosmilus* to Clupeomorpha and Clupeoidei is based on its close relationship with †*Clupeopsis*, as the only known specimen belonging to the former genus does not preserve any anatomical region that would be diagnostic for those clades. We added the following sentence to the Discussion, section 4.b: “Instead, the placement of †*Monosmilus* within these groups is based on strong evidence that it is closely related to †*Clupeopsis* (see previous section).”

Page 9, line 24. Not clear what "This" is referring to. If it refers to the previous sentence, I am not sure how a very fragmentary quadrate is indicative of the orientation of the hyomandibula of which there is only a head fragment is preserved. Please clarify this sentence.

We acknowledge that the description was ambiguous here. To clarify this, we substituted the sentence “This and the small portion of the hyomandibular head that is preserved indicates that the hyomandibula was strongly reclined posteriorly.” with “The anterior extent of the quadrate is constrained by the dentary, and it is apparent that the jaw joint was located far posteriorly. This position, combined with more anterior location of the small portion of the hyomandibular head that is preserved, indicates that the hyomandibula was strongly reclined posteriorly.”

Page 10, line 31. Because most of the diagnostic features that place *Monosmilus* in the Clupeomorpha:Clupeoidea:Engrauloidea are not preserved in the specimen, it would be reasonable to add a paragraph acknowledging this.

See comment referring to page 8, line 14 for the placement of †*Monosmilus* within Clupeomorpha and Clupeoidei. The placement of †*Monosmilus* within Engrauloidea is supported by some features preserved in the specimen (or that can be inferred from the specimen) that are already discussed here: suspensorium posteriorly inclined; substantial portion of metapterygoid anterodorsal to quadrate; absence of basihyal and basihyal toothplate (see comment below).

Page 10, line 50. Toothless basihyal is also found in some non-engraulid clupeiforms... *Nematalosa*, *Konosirus*, *Sardinops*, *Spratelloides*, etc.

The feature we cite here from Nelson (1970) that we observe in †*Monosmilus* and (less clearly) in †*Clupeopsis* is the absence of an ossified basihyal and of a basihyal toothplate. We cannot check the condition seen in all the genera listed in this comment, but *Konosirus* has a basihyal toothplate (even though it is toothless). This is a different condition from what is seen in engraulids and in †*Monosmilus* and †*Clupeopsis*, which either do not have a basihyal toothplate at all or have it extremely reduced.

Page 11, line 36. Considering that chirocentrids have been suggested to be engrauloids based on some morphological features also observed in *Clupeopsis* and *Monosmilus* (DiDario 2009), it becomes a very interesting question to assess phylogenetic relationships of these groups in a cladistic framework...this is just an aside, as I see this particularly interesting.

We expanded upon this by adding these few sentences to section d) of the Discussion: “Our placement of the large-fanged †*Monosmilus* and †*Clupeopsis* as stem engraulids provides a new perspective bearing on the hypothesized relationship between chirocentrids and anchovies. Although these Eocene genera clearly possess many derived features of engraulids that are absent in chirocentrids, the overall structure of mandibular dentition (in both genera) and braincase (in †*Clupeopsis*; incomplete in †*Monosmilus*) is remarkably similar to that of *Chirocentrus*. These features could represent generalized conditions of the putative engraulid/chirocentrid radiation, subsequently lost in more crownward engraulids.”

The following sentence was changed to better fit the revised version of section d), from “Combined with subsequent phylogenetic interpretation of the Early Cretaceous †*Cynoclupea* as sister to the putative chirocentrid/engraulid clade, this led to the hypothesis that suspension feeding engraulids are derived from a piscivorous ancestor” to “The Early Cretaceous †*Cynoclupea* joins *Chirocentrus* as another putative relative of the engraulid clade, leading Malabarba and Di Dario to hypothesize that planktivorous engraulids are derived from a piscivorous ancestor.”